# An externally validated resting-state brain connectivity signature of pain-related learning
Balint Kincses [1,2] ✉, Katarina Forkmann[1], Frederik Schlitt [1], Robert Jan Pawlik[3], Katharina Schmidt [1], Dagmar Timmann[1], Sigrid Elsenbruch[1,3], Katja Wiech [4], Ulrike Bingel [1,5] & Tamas Spisak [1,2,5]

Pain can be conceptualized as a precision signal for reinforcement learning in the brain and alterations in these processes are a hallmark of chronic pain conditions. Investigating individual differences in pain-related learning therefore holds important clinical and translational relevance. Here, we developed and externally validated a novel resting-state brain connectivity-based predictive model of pain-related learning. The pre-registered external validation indicates that the proposed model explains 8-12% of the inter-individual variance in pain-related learning. Model predictions are driven by connections of the amygdala, posterior insula, sensorimotor, frontoparietal, and cerebellar regions, outlining a network commonly described in aversive learning and pain. We propose the resulting model as a robust and highly accessible biomarker candidate for clinical and translational pain research, with promising implications for personalized treatment approaches and with a high potential to advance our understanding of the neural mechanisms of pain-related learning.

Chronic pain exhibits considerable inter-individual variability which has been linked to a combination of physical, neural, psychological, and social factors[1–3]. Recent multivariate studies using brain-based predictive models[4–8] and "neurotraits"[9] hold promise to provide a composite window into pain states and, ultimately, escape the constraints of the case-control diagnostic paradigm by offering a set of objective measures of individual-level pain characteristics[10].

An underexplored avenue for expanding the pool of potential brain-based biomarkers of pain is to target individual differences in pain-related learning processes. Pain and learning are intimately linked, and can be conceptualized within a common framework, with pain being a precision signal for reinforcement learning in the brain[11]. Pavlovian conditioning plays a fundamental role in many aspects of pain behavior[12], including fear and threat detection[13], escape and relief learning[14], and even in placebo analgesia[15]. In acute pain, we learn to associate painful experiences with certain stimuli or situations, which helps us to adapt by learning to avoid or minimize future harm. In persistent pain states, this learning can become maladaptive and a key driver of pain chronification[16,17]. Understanding individual differences in pain-related learning and its underlying neural mechanisms is therefore key not only to a comprehensive understanding of

(chronic) pain but also to the development of effective prevention and treatment options.

Previous structural and functional imaging studies suggest that the underlying neural processes of aversive learning are rooted in a network of brain regions such as the amygdala[18–22], hippocampus[19,23,24], dorsal anterior cingulate cortex[24,25], posterior insula[24,26], cerebellum[24,27], and prefrontal regions[19,24]. Although pain-related aversive learning and pain conditioning clearly use the neural mechanisms of general aversive learning and threat conditioning, recent research also suggests a clear distinction in mechanisms and the presence of networks that are specific to pain-related learning[28,29]. Pain-related learning performance is relatively consistent within an individual over time[30] and the corresponding neural substrates also seem to be reflected in resting brain activity, i.e., in the absence of an explicit task[31–36]. However, it has yet to be determined whether resting-state brain connectivity can characterize and provide a better understanding of individual differences in pain-related learning.

Here we aimed to develop and evaluate a resting-state functional brain connectivity-based multivariate predictive marker of individual differences in pain-related learning, as measured by an established classical conditioning paradigm.

[1]Department of Neurology, Center for Translational Neuro- and Behavioral Sciences, University Medicine Essen, Essen, Germany. [2]Institute for Diagnostic and Interventional Radiology and Neuroradiology, University Medicine Essen, Essen, Germany. [3]Department of Medical Psychology and Medical Sociology, Faculty of Medicine, Ruhr University Bochum, Bochum, Germany. [4]Wellcome Centre for Integrative Neuroimaging, FMRIB, Nuffield Department of Clinical Neurosciences, University of Oxford, Oxford, UK. [5]These authors contributed equally: Ulrike Bingel, Tamas Spisak. ✉e-mail: balint.kincses@uk-essen.de

**Fig. 1 | Functional connectivity model predicts pain-related learning performance.**
**a** Generalizable predictive performance estimates were obtained via a "registered model" design. Our resting-state connectivity-based marker of pain-related learning (RCPL) model was developed in a discovery sample. Prior to estimating the model's true, unbiased performance in an independent external sample, we fixed all model parameters and pre-registered our approach. **b** Behavioral results: in all three datasets (Ds – discovery sample ($n = 25$), Vs1 – validation sample 1 ($n = 26$), Vs2 – validation sample 2 ($n = 23$)), we observed an increase in valence ratings of the pain- and tone-related conditioned stimuli ($CS^+_{pain}$ and $CS^+_{tone}$, respectively), as compared to the safety signal ($CS^-$), indicating successful conditioning. Mean and standard error of the ratings in each experimental phase are depicted by line plots, separately for all three datasets. Individual rating values are visualized (red dot – $CS^-$; green triangle – $CS^+_{pain}$; blue rectangle – $CS^+_{tone}$). **c** Model performance: our model successfully predicted individual differential valence ratings (DVC, see Eq. 1) in the external validation sample ($r = 0.335$, $p = 0.009$). Validation sample 1 ($n = 26$) and 2 ($n = 23$) are denoted by marker shape (validation sample 1 – triangle, validation sample 2 – rectangle).

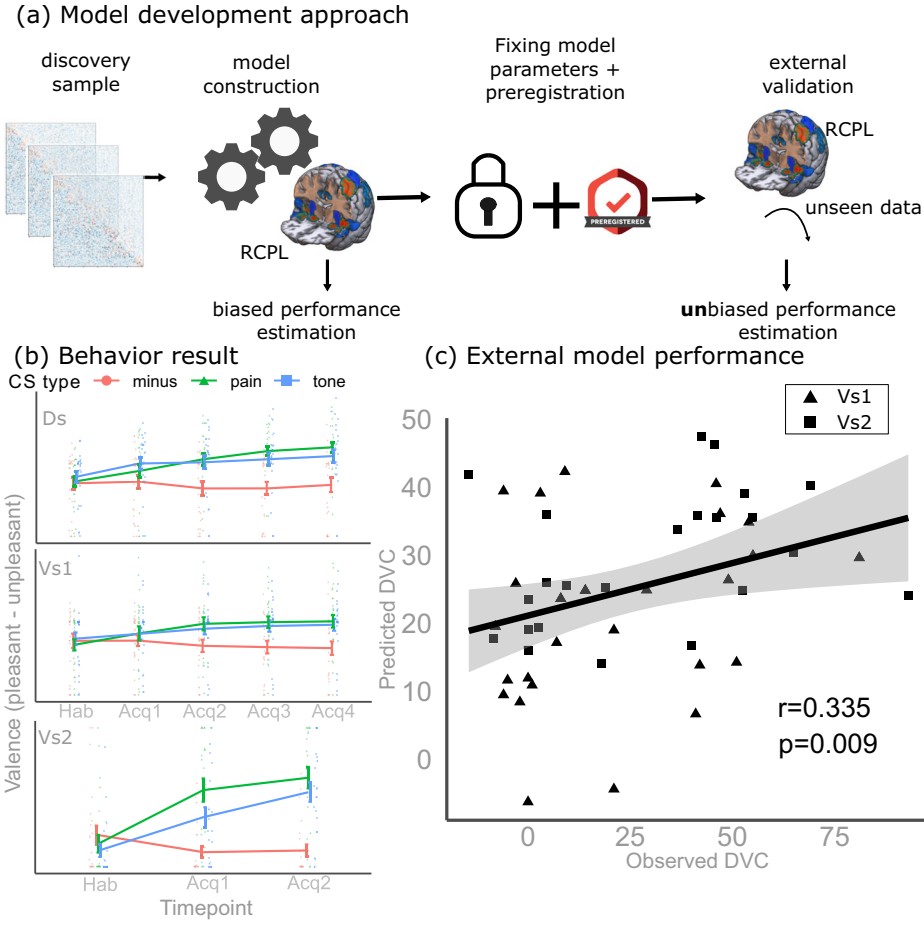

To ensure transparency and reliability, we have followed a so-called "registered model" design[37] that has addressed concerns about overly optimistic performance estimates during model discovery[38–44]. This included (i) limiting analytical flexibility by using a standard preprocessing pipeline and methodological recommendations from the literature, (ii) fixing and publicly registering all model parameters (including data pre-processing) after the model discovery phase, (iii) externally validating the proposed model on prospective and retrospective datasets, to provide a robust and unbiased assessment of predictive performance. Finally, we aimed to evaluate the model in terms of specificity for pain-related learning, bias toward potential confounders, and neuroscientific validity/plausibility of the underlying predictive brain network.

## Results

### Pain and aversive tone stimuli elicited associative learning behavior

We assessed individual learning performance related to pain and aversive tone using a differential conditioning paradigm (Supplementary Fig. 1) in a total of $n = 99$ ($n = 74$, after exclusion) young healthy volunteers (age: $25.3 \pm 3.9$ years (mean ($\pm$sd)) in three separate studies (discovery sample + validation samples 1 and 2) see Supplementary Table 1 for study-wise demographic summaries). In brief, the unconditioned stimuli (US pain and tone) were preceded with geometric figures as conditioned stimulus (CS) and participants rated the unpleasantness of these CS (via valence ratings). Another geometric figure that was never followed by a US served as control ($CS^-$). Over the course of the experiment, participants learned to associate the CSs with the USs which was reflected in the change of valence ratings (see section "Methods" and Fig. 1b for more details).

In all three studies, the behavioral results showed evidence for threat learning with both CS (see Fig. 1b – Behavior results, $p < 0.05$ for all, Table 1, Supplementary Tables 2–5).

Individual pain-related learning performance was characterized by differential valence ratings (see Eq. 1), i.e., the degree to which the valence difference between $CS^+_{pain}$ and $CS^-$ (see Eq. 1) changed over the course of the experiment for a given individual. Differential valence ratings served as the target measure for the discovery and validation of our brain-based predictive model of pain-related learning.

### Resting connectivity-based signature predicted individual pain-related learning

Resting-state functional MRI data were collected from all 99 participants ($n = 74$, after exclusion) in the three studies for a duration of 8–10 min/participant, prior to the behavioral experiments (see Supplementary Table 1). The model development was performed exclusively in the discovery study ($n = 25$, after exclusions), prior to acquiring the validation data. We trained a machine learning model, based on whole-brain resting-state functional brain connectivity, assessed between $M = 122$ functionally defined regions (BASC atlas)[45] to predict pain-related learning performance as quantified by differential valence ratings (Eq. 1, Supplementary Fig. 2). Nested leave-one-participant-out (LOPO) cross-validation indicated a significant predictive performance in the discovery sample (Supplementary Fig. 3). The final regularized linear model (Ridge) was fitted on the discovery sample and contained ten predictive connections with a regularization parameter of $\alpha = 0.001$. However, recent studies[39,44] have raised concerns about the reliability, replicability, and generalizability of effect sizes measured in small-sample studies, particularly when using LOPO cross-validation[38]. Therefore, we evaluated the true, unbiased predictive effect size of our model by means of an independent, pre-registered external validation, based on two, independent, heterogenous samples that were not available during model discovery.

Prior to obtaining the external validation data, we froze all model weights, fixed all preprocessing parameters, and pre-registered our

## Table 1 | Behavioral results

| | Pain | | Aversive tone | |
|---|---|---|---|---|
| | **Valence** $\Delta(\Delta CS^+_{pain}, \Delta CS^-)$ (95% CI) | **Contingency**[a] $CS^+_{pain}$ (95% CI) | **Valence** $\Delta(\Delta CS^+_{tone}, \Delta CS^-)$ (95%CI) | **Contingency**[a] $CS^+_{tone}$ (95% CI) |
| Discovery study (n = 25) | 25 (16–34) | 56 (40–70) | 15 (7–25) | 22 (1–42) |
| Validation study (n = 49) | | | | |
| Sub-sample 1 (n = 26) | 21 (12–31) | 48 (29–65) | 15 (4–25) | 19 (1–37) |
| Sub-sample 2 (n = 23) | 29 (18–41) | 66 (50–80) | 27 (16–38) | 70 (55–84) |

Results show evidence for threat learning in both the pain and the unpleasant tone conditions, in all three studies. Values represent the mean and bootstrapped confidence intervals value in parenthesis. $\Delta CS^+_{pain}$ – valence change between the end of acquisition and habituation for the conditioned stimulus associated with pain, $\Delta CS^+_{tone}$ – valence change between the end of acquisition and habituation for the conditioned stimulus associated with aversive tone, $\Delta CS^-$ – valence change between the end of acquisition and habituation for the conditioned stimulus associated with no aversive stimulus. See Eq. 1 for more details. $CS^+_{pain}$ – contingency ratings of the conditioned stimulus associated with pain, $CS^+_{tone}$ – contingency ratings of the conditioned stimulus associated with aversive tone. A similar table with all the participants (without exclusion) can be found in the Supplementary Material (Supplementary Table 4).
[a]Contingency ratings were measured with different scales in the discovery and validation study 1 as compared to validation study 2 (see the section "Methods", subsection "Behavioral paradigm").

approach (https://osf.io/b8znd, Fig. 1a – Model development approach). We refer to this model as the Resting-state functional Connectivity signature of Pain-related Learning (RCPL).

The external validation involved no model fitting. Instead, we simply obtained the RCPL predictions independently for each participant and contrasted them to the behavioral results, pooled across all participants in the external validation sample.

In this independent data, we observed a highly significant predictive performance ($r = 0.34$ $CI_{90\%}$: [0.14 0.53], $p = 0.009$, Fig. 1c – External model performance). The observed effect size estimates were consequent in both sub-samples, ($r = 0.28$, $CI_{90\%}$: [−0.01 0.54] and $r = 0.35$, $CI_{90\%}$: [0.04 0.68] in validation sample 1 and 2, respectively) showing that the RCPL model can be expected to explain 8–12% of variance in independent datasets (based in validation study 1 and 2, respectively; see Supplementary Fig. 3 – External samples). The root mean squared error values were 26.8, 25.03, and 26.9 for the merged sample, validation sample 1, and validation sample 2, respectively. The range of the observed values was [−14.5, 98.0] within the validation dataset. Interestingly, completely dropping our pre-registered motion-exclusion (~20% more individuals), the performance of the model remained unchanged ($r = 0.35$, $p = 0.0028$, see also Supplementary Notes: Model performance on the extended sample, Supplementary Fig. 4).

### Specificity to pain-related learning and confounding bias

To be considered as a valuable tool for basic and translational clinical research, our findings on the externally validated, out-of-sample predictive performance of our proposed model must be complemented by an initial investigation of the model's convergent and divergent validity[41]. Here we aimed to characterize the model's (i) generalizability to other measures of pain-related learning (convergent validity to contingency ratings and extinction learning), (ii) its specificity to pain (divergent validity to tone-related learning), and (iii) confounding bias (divergent validity to various confounding variables, such as fear of pain or in-scanner motion artifacts).

Our criteria for generalizability, specificity, and confounding go beyond the common practice of drawing conclusions simply from bivariate associations between the model predictions and the (convergent and divergent) validator variables. Instead of this arguably misleading approach, we focus on the conditional independence structure across all three variables involved (i.e., target, prediction, validator) with a dedicated statistical test[46]. In our definition, predictions are specific to the target variable, if they are conditionally independent of the validator variable, i.e., their (marginal) association is not stronger than would be expected from the association between the target and the validator variable. Conversely, if the predictions are conditionally dependent on the validator variable (i.e., the validator explains variance in the predictions over and beyond the expected marginal association), we say that the predictions generalize to the validator variable. In our view, confounding bias is equivalent to an (unwanted) generalization to a divergent validator (for more details on our approach see ref. 46). The concepts of convergent and divergent validity are not yet fully developed in the field of machine learning. Importantly, the term generalization to other

measures of pain (or convergent validity) should not be confused with the model generalization to new, unseen data (i.e., external validity see above). Additionally, our specificity analysis (divergent validity) tests whether the model captures aversive learning (tone-related) in general and differs from the similar term used in classification problems.

Differences between specificity assessment with bivariate measures and conditional independence can be illustrated by testing the convergent validity of our model with another measure of pain-related learning such as contingency rating. In contrast to the affective aspect of learning (measured by the change in valence ratings, Eq. 1), contingency captures the cognitive component of pain-related learning. While the relationship between contingency learning and the prediction of our model was significant ($R^2 = 32\%$, $p = 0.003$), we also have to take into account that contingency learning and valence learning were also highly correlated ($R^2 = 17\%$, $p = 0.038$). Therefore, to demonstrate that the model generalizes to contingency learning, we need to reject the null hypothesis of conditional independence between the prediction and contingency learning (given the association of both with differential valence rating). The specific statistical test was able to reject this null hypothesis with $p = 0.05$, meaning that the association between the prediction and contingency learning was not only significantly different from zero, but it was significantly higher than expected from the association between valence learning and contingency learning (Fig. 2 – Convergent validity, Table 2, see also Supplementary Notes: Model validators in the validation sample, Supplementary Table 6).

Similarly, the predictions of the RCPL signature were significantly associated with differential valence ratings (internally validated estimate: $R^2 = 51\%$, $p < 0.001$, nested LOPO cross-validation) not only during the acquisition phase but also during extinction ($R^2 = 18.6\%$, $p = 0.03$) (i.e., the recovery of valence ratings after cessation of conditioning ($CS^+$–US pairing)). However, when testing for conditional independence between extinction and the model predictions, we found no evidence that this association was more than a consequence of the correlation between acquisition and extinction learning alone ($R^2 = 46\%$, $p = 0.86$; Fig. 2 – Convergent validity, Table 2, see also Supplementary Notes: Model validators in the validation sample, Supplementary Table 6).

The model predictions were also significantly correlated with valence learning induced by the aversive tone ($R^2 = 18.9\%$, $p = 0.03$). However, the specific statistical test suggested that this association could be explained as a secondary consequence of the association between pain-related learning and learning induced by the aversive tone ($R^2 = 49\%$). Thus, our analysis could not provide evidence against the specificity of our model for pain-related learning. Note that if our model would primarily capture general aversive learning (i.e., neural mechanisms shared between pain and tone learning), its predictions would also be in conditional dependence with tone learning (Fig. 2 – Divergent validity, Table 2, see also Supplementary Notes: Model validators in the validation sample, Supplementary Table 6).

In all three datasets, we have obtained a comprehensive set of additional demographic and psychometric measures as well as measures of in-scanner motion (see all the variables in Supplementary Table 3). The partial

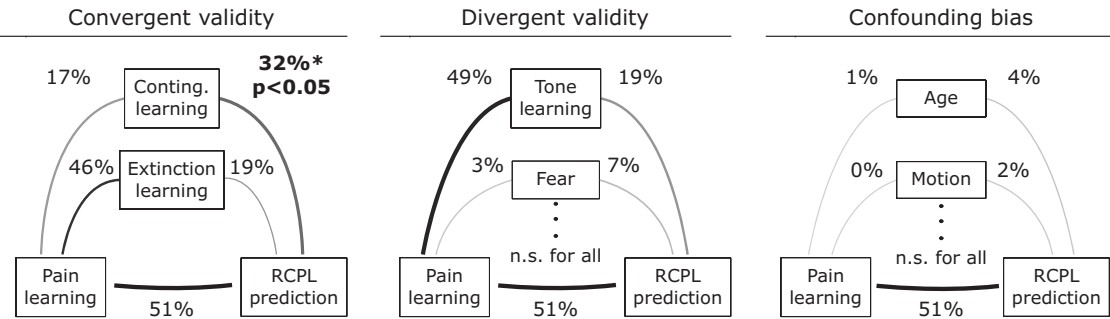

**Fig. 2 | Analysis of convergent validity, divergent validity, and confounder bias.** Convergent validity: testing the generalizability of the model to the cognitive aspect of learning suggests that the RCPL signature captures pain-specific learning, as reflected in the rejection of conditional independence of contingency ratings ($p = 0.05$). However, the model prediction has not generalized to extinction learning ($p > 0.05$). Divergent validity: model prediction was tested against tone-related aversive learning. The model was not driven by tone learning, suggesting that the model primarily captures pain-related learning and not tone. Other divergent validators were also tested, including fear of pain/tone, pain catastrophizing, and pain anxiety, but none of them significantly biased the model. Confounding bias: none of the investigated confounder variables including age, in-scanner motion parameters (mean, median and maximum framewise displacement, percent of scrubbed volumes), and depression score biased our model (see Supplementary Table 3). The thickness and the opacity of the lines are proportional to the $R^2$ values. Percentages refer to the coefficient of determination $R^2$ between the two variables. $p < 0.05$ indicates that the model prediction is significantly associated with the variable over and above the baseline association between the observed pain-related differential valence ratings (DVC, Eq. 1) and the variable itself (e.g., contingency learning). The observed and predicted valence learnings were significantly correlated $R^2 = 51\%$, $p < 0.001$. Conting. Learning – $CS^+_{pain}$ contingency after acquisition training, Extinction learning – differential valence rating in extinction learning (see the study preregistration for more detail), Tone learning– differential valence rating for tone (see Eq. 1), Fear – fear of pain (see "Questionnaires"), Motion – head motion during the resting-state fMRI (mean framewise displacement (FD)).

## Table 2 | Result of partial generalization test

| | | $R^2$ with pain learning (%) | $R^2$ with model prediction (%) | Significance of partial generalization test (*p*) |
|---|---|---|---|---|
| Convergent validity | Contingency $CS^+_{pain,acq}$ | 17.4 | 31.9 | **0.05*** |
| | Contingency $CS^+_{pain,ext}$ | 1.4 | 25.3 | **0.03*** |
| | Extinction learning$_{pain}$ | 46.1 | 18.6 | 0.86 |
| Divergent validity | Tone learning | 49.1 | 18.9 | 0.80 |
| | Extinction learning$_{tone}$ | 33.5 | 7.5 | 0.92 |
| | Fear of pain | 2.9 | 6.7 | 0.30 |
| | Fear of tone | 0.5 | 0.0 | 0.96 |
| | Pain catastrophizing | 0.0 | 0.0 | 0.91 |
| Pain anxiety | PASS D1 | 0.3 | 0.0 | 0.99 |
| | PASS D2 | 6.5 | 0.8 | 0.78 |
| | PASS D3 | 2.0 | 1.1 | 0.64 |
| | PASS D4 | 0.4 | 1.1 | 0.62 |
| Confounder bias | Age | 1.3 | 3.5 | 0.41 |
| Motion | Mean FD | 0 | 2.0 | 0.49 |
| | Median FD | 0.2 | 1.8 | 0.52 |
| | Max FD | 2.5 | 7.5 | 0.24 |
| | Percent scrubbed | 0.2 | 1.0 | 0.63 |
| | Depression | 0.5 | 0.7 | 0.69 |

We tested convergent and divergent validators, and confounder variables for model prediction. See the main text for a detailed description of convergent, divergent validators, and confounder bias. Significant results in the partial generalization test are denoted with an asterisk and bold character.

confounder test[46] did not provide evidence for confounding bias with any of these variables (Fig. 2 – Confounding bias).

While our power analysis (Supplementary Fig. 5) suggests that larger samples are needed to reliably disentangle general and pain-specific mechanisms in aversive learning, our results provide a promising basis for further large-scale efforts to clarify the divergent validity of the proposed marker.

### Pain-related learning was predicted by a neuroscientifically plausible neural system

Our machine learning procedure involved a data-driven feature selection step and identified 10 predictive connections (out of 7503), 5 of them being positively and 5 negatively associated with pain-related learning (see Table 3 and Fig. 3a). As previous studies report that, even for robust predictive models, model coefficients can be instable and unreliable[47], we did not consider these as the primary measure for quantifying predictive importance. Instead, we have constructed a more reliable, externally validated importance measure of single connections and regions by repeating the whole model fitting procedure in the discovery dataset with a reduced feature space that excludes (i) a single connection or (ii) all connections of a single brain region. Predictive importance is then quantified as the loss of explained variance when testing these "truncated" models in the external validation samples (hereinafter referred to as: "exclusion loss"). As compared to the full model, larger exclusion loss indicates higher predictive importance of the left-out connection/region.

With this approach, the following connections were found to be most predictive to pain-related learning: amygdala – parieto-occipital associative area (exclusion loss: 10.68% of variance explained), posterior insula – left sensorimotor cortex (exclusion loss: 10.33%), angular gyrus – right anterior PFC (exclusion loss: 9.3%) and cerebellum (I–V) – posterior cingulum (exclusion loss: 5.81%) (Fig. 3c). In terms of regional importance (i.e., when excluding all connections of a brain region), the posterior cingulum (exclusion loss: 11.07%), the amygdala (exclusion loss: 9.98%), the posterior insula (exclusion loss: 9.44%), the dorsal PCC (exclusion loss: 7.65%), and the left sensorimotor region (exclusion loss: 7.84%) were found to be the most important predictors (Fig. 3b).

## Discussion

Here we present the RCPL signature, a novel resting-state functional brain connectivity-based predictive model of individual pain-related learning

**Table 3 | Predictive connections of RCPL signature**

| ß coef value | Region 1 | | | | | | Region 2 | | | | | | LOFO (%) |
|---|---|---|---|---|---|---|---|---|---|---|---|---|---|
| | Region name | BASC idx | LORO (%) | x | y | z | Region name | BASC idx | LORO (%) | x | y | z | |
| −79.3 | ParOcc assoc | 100 | 5.9 | 17.13 | −76.26 | 25.61 | Amy | 71 | 9.98 | 3.06 | −7.47 | −19.9 | **10.68*** |
| 84.4 | pIns | 21 | 9.44 | −2.01 | −7.26 | −3.49 | SensMot | 8 | 7.84 | −40.99 | −22.62 | 61.9 | **10.33*** |
| −92.7 | Angular | 50 | 5.56 | −29.83 | −72.86 | 31.94 | r aPFC | 92 | 6.87 | 42.57 | 42.79 | 10.85 | **9.3*** |
| −80.8 | Cer V | 48 | 2.99 | −0.4 | −46.11 | −23.17 | pCing | 19 | 11.07 | −1.47 | −63.33 | 8.28 | **5.81*** |
| 43.3 | Cer VII | 18 | −0.09 | 1.9 | −79.54 | −48.01 | Coll s | 79 | 0.15 | −4.4 | −41.99 | −12.66 | 1.57 |
| 48.4 | iParietal | 114 | −0.45 | −48.23 | −65.34 | 33.46 | Marg s | 118 | −1.71 | −0.23 | −30.3 | 47.49 | 0.77 |
| −30.5 | aIns | 85 | 2.53 | 10.94 | 22.89 | 4.96 | Vis assoc | 43 | 2.35 | 12.89 | −83.54 | 18.77 | −0.12 |
| 6.8 | aPFC | 87 | 2.21 | 1.64 | 62.77 | 13.2 | STG | 88 | 6.44 | 0.66 | −5.24 | −0.93 | −0.63 |
| −49.2 | Put | 25 | 6.03 | −2.8 | −2.14 | 4.11 | Vis assoc | 16 | −2.66 | 2.23 | −95.23 | 14.76 | −2.21 |
| 38.6 | dPCC | 72 | 7.65 | −1.71 | −49.05 | 7.1 | sParietal | 76 | −5.45 | 0.57 | −25.7 | 73.98 | −2.29 |

The table lists all predictive connections in the RCPL signature, including the ß coefficient of the connections in the model (the positive (negative) sign of the ß coefficient means that a higher connection between the two regions leads to higher (lower) differential valence rating), the involved regions, their coordinates, their index in the applied functional brain atlas (generated with the bootstrap analysis of stable clusters method (BASC)), and the exclusion loss in the leave-one-out-region (LORO) and leave-one-out-feature (LOFO) analysis. The table is sorted based on feature importance as measured by exclusion loss (the loss of explained variance in the external sample when excluding the feature during model construction). Significant drop in explained variance in the LOFO analysis is denoted with asterisk and bold character.

*ß coef* coefficient of ß value in the model, *Region name* custom-defined name of the regions, *BASC idx* the index of the region from the applied BASC atlas, *LORO* exclusion loss in the leave-one-region-out approach (see section "Methods"), high values mean large drop in explained variance compared to the original model – x, y, z: "Center of Mass. We report the center of mass of the parcel in MNI coordinates. Note that the center of mass may lie outside the parcel for bilateral parcels[45]"; Supplementary Table 7 contains MNI coordinates of left and right parcels separately for bilateral parcels, *LOFO* exclusion loss in the leave-one-feature-approach (see section "Methods"), high values mean larger drop in explained variance, *ParOcc* parieto occipital association area, *Amy* amygdala, *pIns* posterior insula, *SensMot* sensorimotor area, *Angular* angular region, *r aPFC* right anterior prefrontal cortex, *Cer V* cerebellar lobules I–V, *pCing* posterior cingulum, *Cer VII* cerebellar lobule VII, *Coll s* collateral sulcus, *iParietal* inferior parietal lobule, *Marg s* marginal sulcus, *aIns* anterior insula, *Vis assoc* visual associative area, *aPFC* anterior prefrontal cortex, *STG* superior temporal gyrus, *Put* putamen, *dPCC* dorsal posterior cingulate cortex, *sParietal* superior parietal lobule.

*Significant drop in explained variance.

performance. We discuss the relevance of the presented brain biomarker of pain-related learning for future research efforts and clinical applications, as well as the proposed model's externally validated predictive performance, specificity, and its neurobiological plausibility.

The RCPL signature offers a wealth of opportunities to advance clinical and translational pain research, both as a stand-alone tool to quantify pain-related learning, and as a potential component of future composite pain biomarkers[10].

First, RCPL scores can be interpreted as the signature of an underlying neural mechanism that may be differentially expressed in different pain conditions. Using this signature as an easily quantifiable indicator could help to identify those at risk of chronification due to (maladaptive) pain-related learning.

Second, our brain signature may allow for a more in-depth mechanisms-based characterization of patients presenting with similar pain phenotypes. As with structural connections, the expression of functional links between brain regions varies widely between individuals, determining the extent to which they learn. Insights into the strength and integrity of these connections could therefore provide valuable information about an individual's capability for pain-related learning and serve as both a prognostic and diagnostic tool.

Third, the RCPL signature could be used to assess or predict treatment efficacy, as well as a method for patient stratification. Combined with improvements in subjective outcome measures, changes in the RCPL signature could indicate whether an intervention has effectively altered pain-related learning (as opposed to other, less relevant processes). As the RCPL signature is derived solely from resting-state functional connectivity, and therefore requires only a 10-min non-invasive scan without the use of noxious stimuli or time-consuming learning paradigms, it is a highly accessible tool for these clinical and translational purposes.

While the development of pain-related brain biomarkers has been widely hailed as an important step[10], a thorough evaluation of their predictive performance, specificity, and confounders is warranted. In our pre-registered external validation, the RCPL signature was found to explain 8–12% of the variance in pain-related learning. This is a substantial predictive effect size, both according to Cohen's rule of thumb[48] and as compared to typical effect sizes in multivariate brain-wide association studies. For example, 90% of the psychometric traits in the Adolescent Brain Cognitive Development dataset show weaker predictability and the proposed model is on par with the top 10% phenotypes in the same dataset[43].

While large effect sizes are desirable for a biomarker for individual predictions in clinical settings, effect sizes that are comparable to, or considerably lower than the hereby reported effect size, proved to be useful in the case of polygenic scores[49–51]. For example, in studies assessing inter-individual variance in cognitive performance, genetic data typically explain around 7–10% of the variance, even with data from more than a million individuals[52]. With effect sizes, similar to that of the proposed model, polygenic risk scores are commonly deployed as research tools to explore the genetic architecture of complex traits and diseases, to uncover new genetic associations, to identify novel pathways, and to gain a deeper understanding of the underlying biology[53]. Similarly, investigating the "expression" of the identified brain network signature of pain-related learning in various experimental and clinical pain conditions provides a promising avenue for further research, with the potential to inform the development of novel, personalized treatment approaches.

Of note, we observed a remarkable difference between internally and externally validated estimates of prediction performance. The reasons may be twofold. First, in small-sample model discovery, cross-validation is known to provide unbiased, but highly variable estimates[38,39]. Second, in the case of complex preprocessing and feature engineering workflows, it is often not feasible to include all free parameters of the model in the nested cross-validation procedure[37,38]. Seemingly "innocent" adjustments of such workflows during model discovery can lead to biased internal performance estimates. Our study shows that both pitfalls can be overcome with pre-registered external validation; a technique that should become standard practice, in order to unlock small-sample model discovery and accelerate the discovery of robust and replicable multivariate brain-behavior associations[43,44].

The model was trained to predict pain-related valence learning. However, that does not automatically imply that the model is specific to pain-related valence learning as the predictions may be driven by more

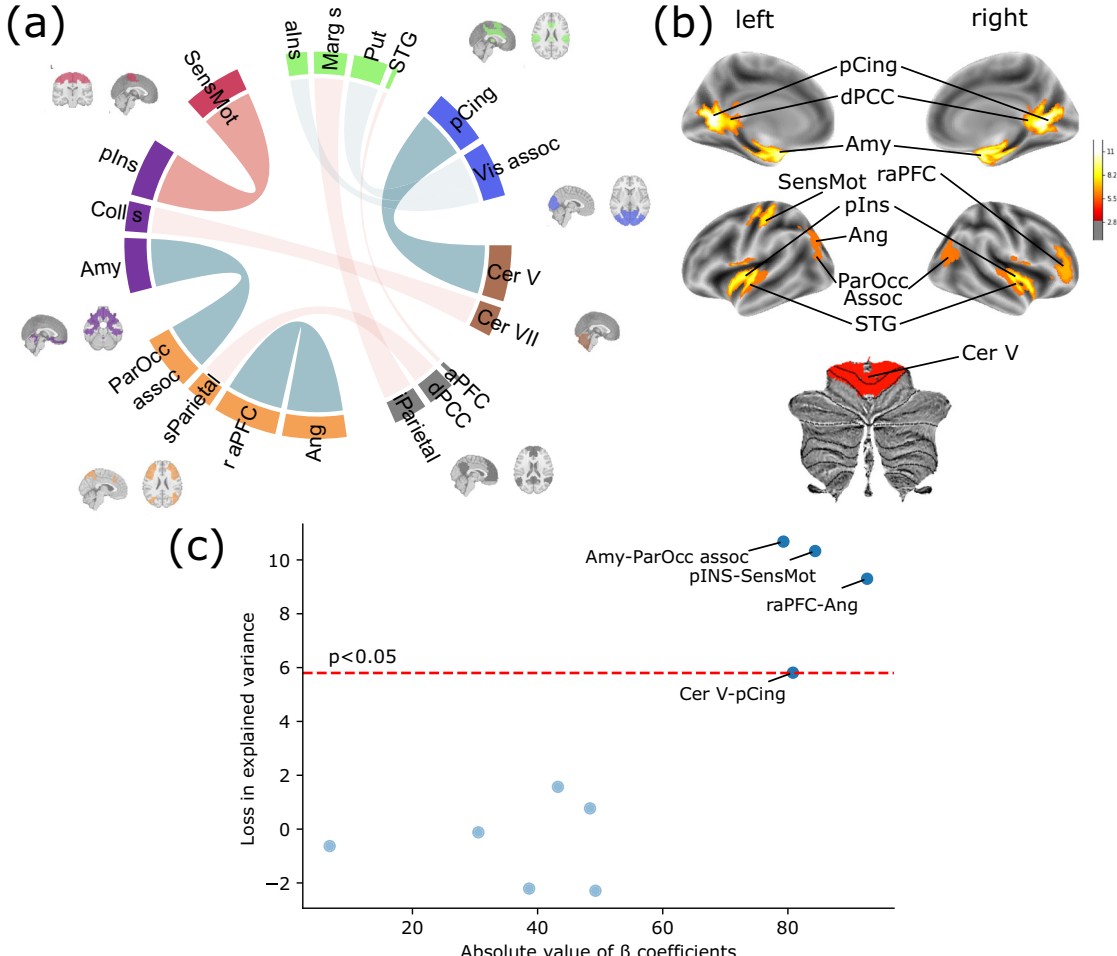

**Fig. 3 | Resting-state functional connectivity predictors of pain-related learning. a** Chord plot of all 10 predictive connections between 20 regions. The sign of the ß-coefficient is color coded, red (blue) indicates a positive (negative) sign, i.e., higher connections between the two regions lead to higher (lower) differential valence rating. The thickness of the chords is proportional to the absolute magnitude of the ß-coefficient. Regions are grouped according to their parent network based on the seven parcellations of the BASC atlas. Red – somatomotor network; green – ventral attention + salience network + basal ganglia + thalamus; blue – visual network; brown – cerebellar network; gray – default mode network; orange – fronto-parietal network + visual downstream; purple – mesolimbic network. **b** Exclusion loss after leave-one-region-out (LORO). Leaving out some regions, such as the amygdala or posterior insula leads to a large decrease in the explained variance of the model when tested in the external validation datasets. Only cortical regions with significant ($p < 0.05$) exclusion loss are visualized. **c** Contrasting the association between the predictive connections' ß-coefficients (the standard technique for assessing predictive performance) and exclusion loss (the proposed more robust measure of importance) in the leave-one-feature-out analysis, shows that six out of ten predictive connectivities have little or no effect on the external model performance. However, the four most important connections, on the other hand, clearly stand out from the rest; excluding them during model training results in models that explain 6–10% less variance in the external samples. The red horizontal dashed line represents the significance threshold for the exclusion loss (5.8%). Abbreviations of the regions can be found in the footnote of Table 3.

general aspects of aversive learning, with shared underlying mechanisms[28,54,55], that generalize across modalities (e.g., aversive tone-related learning) and across emotional and cognitive aspects (e.g., contingency ratings). Therefore, the specificity of the model was additionally tested against these other aspects of aversive learning. We found that the RCPL does significantly generalize to (i.e., explained more variance than expected in, convergent validity) the cognitive aspect of pain-related learning (contingency ratings). On the other hand, we found no evidence for "generalization" to aversive tone-related learning (divergent validity).

The absence of bias in the predictions toward potential confounding variables is also a critical factor in determining the clinical relevance and translational potential of brain-based predictive models[46]. For instance, psychological factors such as anxiety and depression are known to be associated with aversive learning[56]. In MRI-based predictive models, motion artifacts are also known to be problematic, especially when the variable of interest is also correlated with motion in the scanner[57,58]. Our confounder analysis found no evidence that the predictions of the proposed model are biased by variables of no interest, such as in-scanner motion, demographics, or depression, making the RCPL signature as a promising candidate for further independent validation.

Using a machine learning approach with inherent feature selection functionality, we identified a set of ten resting-state connections that together predicted pain-related learning (Table 3). Five of these connections were positively associated with learning, while the remaining five were negatively associated. As recent work has highlighted potential reliability issues with feature weights from predictive models[47], we retrained models with feature exclusion to obtain more reliable, externally validated importance estimates. We found that 4 of the 10 connections and 9 out of the 20 involved brain areas had statistically significant external validity. Most of the identified predictors are known to play a critical role in aversive learning, pain-related neural processes, or both.

First, mesolimbic regions, and the amygdala in particular, make the largest contribution to the predictive capabilities of the RCPL signature. Connectivity between the amygdala and the frontal part of the central executive network

(CEN) (which is also known as frontoparietal network[59]) is one of the most prevalent findings in aversive learning and is commonly associated with behavioral learning parameters (US expectancy, fear ratings)[31,33]. Although, unlike our study, these studies examined resting-state connectivity changes induced by learning (as opposed to how baseline connectivity shapes subsequent learning performance), they confirm the important role of these regions in the underlying neural processes. Our results substantially contribute to these findings by highlighting the critical role of the connection between the amygdala and the parietal part of the CEN (occipitoparietal association area), a network widely recognized for its involvement in pain and perception more generally[59]. The amygdala has been reported to directly mediate aspects of fear learning and to facilitate fear memory operations in other regions, including the hippocampus and prefrontal cortex[60]. Amygdala–CEN connectivity has been found to be increased in patients with chronic back pain[61], in whom the evaluative aspect of threat learning has been reported to be impaired[62].

Second, the strong contribution of posterior insula-sensorimotor connection to the model may be a key component of our model's apparent specificity for pain-related learning. Modality-specific threat cues have been reported to have shared and distinct biological components[28,29] and there is recent evidence for the specificity of the posterior insula for pain-related learning as compared to tone[63]. The insula has long been proposed as a multisensory integration site in pain[64,65]. Our study extends these findings and suggests that, in addition to the well-documented role of the insula in processing acute pain, pain-related learning features are also encoded in the interplay between sensory and posterior insular regions during spontaneous resting-state activity.

Third, we found that connectivity between the anterior prefrontal cortex and the angular region was negatively associated with pain-related learning. The anterior prefrontal cortex is part of the CEN and has been implicated in various aspects of pain processing[59]. Among other processes, it has been reported to predict cognitive modulation of pain[66], and anterior prefrontal and angular regions (very similar to our findings) were found to be deactivated during fear conditioning[24] and activated during placebo analgesia in a large individual participant meta-analysis[67].

Finally, the cerebellum is known to be involved in fear conditioning behavior[68–70] and there is now also accumulating evidence for a direct involvement of the cerebellum in pain modulation[67,71]. However, the cerebellum is often overlooked in related research. Here we found that omitting cerebellar connectivity significantly reduced predictive performance (~6% less variance explained). These results provide additional evidence that the commonly reported pain- and learning-associated cerebellar activity is not a secondary consequence of the experimental paradigms in these studies (e.g., suppression of motor responses) but it has a unique predictive contribution to the underlying mechanisms that is not redundant with activity and connectivity in other parts of the brain.

While the involvement of these connections in pain-related learning is plausible, further studies are needed to elucidate their specific functional significance. For example, given that both the posterior insula and the sensorimotor area have primarily been implicated in the processing of sensory rather than cognitive-affective information[72,73], their contribution to the RCPL signature suggests a prominent role for sensory processing in pain-related learning. Whether this is indeed the case, needs to be investigated in further studies.

Moreover, the hippocampal formation, that consistently shows increased signal levels during pain-related learning, is represented only by the collateral sulcus (a region tightly linked to the hippocampus[74]), and analysis of its importance in the external validation sample was inconclusive. Therefore, further investigation of its contribution as a unitary region versus as part of an extended network of functionally connected brain areas is needed for a more detailed understanding of its role.

The low reliability of self-reported outcome measures can be an important limitation in terms of the power and reliability of brain-based predictive models[75,76]. This may also apply to the model presented here, which was trained to predict pain-related learning indexed as differential valence ratings, i.e., based on subjective self-reports. Thus, the reported predictive performance is likely to be lower than what could optimally be achieved with more reliable target measures.

Our study used a small discovery sample combined with pre-registered external validation. While this approach allows for an efficient and reliable construction of brain-based predictive models[37], our power calculation (Supplementary Fig. 5) indicated that moderate to large samples[44] will be needed to thoroughly characterize the model in terms of its specificity to pain-related mechanisms, its relationship to other aspects of pain-related learning (e.g., resistance to extinction), and its validity in clinical populations.

Disentangling the common and distinct mechanisms of pain- and tone-related learning is further limited in the present study by their potential interactions in our paradigm, through transfer learning and cross-modality generalization.

Like any fMRI-based study, our study is also limited by low SNR in regions prone to artifacts (e.g., susceptibility artifacts), including regions that are known to be involved in pain-related learning, such as the orbitofrontal cortex. Results relating to these regions should therefore be interpreted with caution.

Beyond the presented neural correlates of individual differences in pain-related learning, the RCPL signature holds potential as a reliable research tool for objectively characterizing such differences in future studies. By employing the RCPL signature in large and diverse populations, including individuals with different chronic pain conditions, we can deepen our understanding of the role of learning processes across different contexts. To facilitate the broader utilization of the RCPL signature in translational and clinical investigations, we offer a containerized implementation of our model. This implementation allows for predictions of individual pain-related learning performance from any resting-state fMRI dataset in a single step (https://github.com/kincsesbalint/paintone_rsn)[77].

In this study, we developed and externally validated a novel brain connectivity-based biomarker candidate for individual differences in pain-related learning. Using an unbiased performance evaluation approach, we demonstrate substantial predictive performance of the proposed RCPL signature with potential clinical relevance. Initial assessments of convergent and divergent validity indicate that the model may generalize to different measures of pain-related learning, but not to aversive tone, fear of pain, or other potential confounders. The model's predictions are based on predictors with high neurobiological plausibility in light of previous studies. Together, this makes the presented model a robust and highly accessible biomarker candidate for clinical and translational pain research, with promising implications for personalized treatment approaches.

## Methods
### Study design
We aimed to train a predictive model based on individual resting-state connectivity which predicts individual differences in a subsequent differential conditioning paradigm. To address previously reported[38] challenges in terms of reproducibility, creditability, and robustness of predictive models, the project was preregistered (https://osf.io/b8znd/) after the discovery study was conducted but prior to the collection of the validation studies (see Participants and samples). The preregistration did not only fix the details of the MRI preprocessing pipeline, the limits of in-scanner motion parameters for exclusion, the differential conditioning paradigm, and the preprocessing of the behavior data (see below) for validation sample 1 but also the results of the model discovery phase (hyperparameter values, model coefficients, model pipeline). For the complete document and full details, visit the study OSF page (https://osf.io/b8znd/).

### Participants and samples
All three studies involved young healthy individuals with no known neurological or psychiatric conditions. Exclusion was based on pre-registered criteria, such as missing behavior or MRI data, incidental findings in MRI, falling asleep during the resting-state measurement (self-report), and in-scanner motion limits (see "MRI preprocessing"). For additional details on

exclusion see the preregistration (https://osf.io/b8znd/) and the previously published manuscripts on the behavior and task fMRI results[29,78,79]. In all studies, the resting-state paradigm preceded the behavior paradigm.

**Discovery sample.** Thirty-eight participants were measured with a resting-state paradigm before they subsequently performed the behavior paradigm (see "Behavior paradigm") inside the scanner. After exclusion, we used data from 25 participants for model training.

**Validation sample 1.** Thirty-three participants were first measured with a resting-state paradigm and the participants performed the behavior paradigm on a separate day out-of-scanner (median 3 days later, range: 1–5 days) (see Supplementary Table 1). After exclusion, 26 participants' data were used.

**Validation sample 2.** An independent study group shared 28 participants' data. This study (study 3) was acquired by different personnel and the participants performed a different paradigm in-scanner right after the resting-state paradigm (see "Behavioral paradigm"). After exclusion, 23 participants' data were used.

The behavior and task-related fMRI data from the discovery sample and validation sample 2 have been previously published[29,78,79]. Validation sample 1 has not been published before.

The study was conducted in accordance with the Declaration of Helsinki and had been approved by the local Ethics Committee (University of Duisburg-Essen, Germany). All participants provided written informed consent. All ethical regulations relevant to human research participants were followed.

### MRI acquisition
All three studies were conducted using a 3T Siemens MAGNETOM Skyra MRI scanner. A high-resolution anatomical image (MPRAGE sequence, parameters in the discovery study and validation sample 1: repetition time (TR): 2300 ms, echo time (TE): 2.07 ms, inversion time (IR): 900 ms, flip angle (FA): 9°, resolution $1 \times 1 \times 1$ mm; validation sample 2: TR: 1900 ms, TE: 2.13 ms, IR: 900 ms, FA: 9°, resolution: $1 \times 1 \times 1$ mm) and an open-eye resting-state functional image (rs-fMRI) were collected with an EPI sequence (discovery study and validation sample 1: 38 slices, resolution $2.5 \times 2.5 \times 3$ mm, TE: 28.0 ms, TR: 2300 ms, GRAPPA: 2, distance factor 15%, 260 volumes; validation sample 2: 46 slices, $3 \times 3 \times 3$ mm, TE: 30 ms, TR: 2500 ms, GRAPPA: 3, 192 volumes) (Supplementary Table 1).

### Behavioral paradigm
All participants performed a differential conditioning paradigm including habituation, an acquisition training phase, and an extinction training phase.

**Discovery sample.** The details of the task are described in the preregistration (https://osf.io/b8znd/) and in ref. 78. Most importantly, geometric figures were used as CS to predict the delivery ($CS^+_{pain}$ or $CS^+_{tone}$) or the absence ($CS^-$) of two different US ($US_{pain}$ and $US_{tone}$). In the habituation phase, the CS and US were presented/delivered once with a duration of 9 and 2.5 s, respectively. During the acquisition phase, 12 out of 16 $CS^+$ were followed by a US (75% reinforcement rate) (Supplementary Fig. 1). In the reinforced trials, 8 s after CS presentation the US was delivered (1 s overlap). In the extinction phase, 12 CS of each condition were presented but no US were applied. The intertrial interval ranged between 6 and 11 s. Heat pain stimuli and unpleasantness-matched auditory stimuli (see "Stimuli and matching procedure" below) were individually calibrated and served as US ($US_{pain}$ and $US_{tone}$). The participants were informed about the possible association between the CS and US. We measured the following parameters on a 0–100 visual analog scale (VAS): valence ratings (verbal anchors: 0 = "very pleasant", 50 = "neutral", 100 = "very unpleasant", once during habituation, four times during acquisition, and three times during extinction), contingency ratings after acquisition and extinction (verbal anchors: 0 = "100%

auditory stimulus", 50 = "0%", 100 = "100% painful stimulus"), US unpleasantness ratings (verbal anchors: 0 = "not unpleasant at all" and 100 = "unbearably unpleasant", once during habituation and four times during acquisition), as well as fear of pain and tone before the experiment. The extinction phase followed the acquisition phase, in which the CSs were not associated with any US.

**Validation sample 1.** The same behavior paradigm was used as in the discovery sample.

**Validation sample 2.** Participants underwent a similar differential conditioning paradigm with two US (75% reinforcement rate), receiving visceral instead of somatic heat pain (Supplementary Table 1). The details of the paradigm can be found in refs. 29,79. During acquisition, 9 out of 12 CS were paired with US. In the reinforced trials, the 14 s long US was preceded by a CS for 6–12 s and terminated together. The intertrial interval was 8 s long. Participants provided CS valence ratings before, in the middle, and after acquisition on a 0–100 VAS (verbal anchors: "not unpleasant at all" and "very unpleasant"). As the anchors of the used scale differed from the one in the discovery study, we corrected for that (dividing the raw values by 2 and adding 50 to them), therefore, the two scales are comparable. Contingency ratings were assessed with a 0–100 VAS (representing the probability of the CS is followed by the specific US).

We derived the pre-registered differential valence change (DVC) from the valence ratings (Eq. 1) in which each element stands for the following: $CS^+_{pain,Acq}$ – is the valence rating of the CS predicting pain given at the end of acquisition training; $CS^+_{pain,Hab}$ – is the valence rating of the CS pain given in the habituation phase (before acquisition training); $CS^-_{Acq}$ – is the valence rating of the CS predicting US omission (not associated with any US) given at the end of acquisition training; $CS^-_{Hab}$ – is the valence rating of the CS predicting US omission (not associated with any US) given in the habituation phase (before acquisition). DVC in the experiment is meant to show the evaluative aspect of learning. A higher value refers to higher learning. We used this value as our main outcome.

$$\text{differential valence change} = \left( CS^+_{pain,Acq} - CS^+_{pain,Hab} \right) - \left( CS^-_{Acq} - CS^-_{Hab} \right) \quad (1)$$

### Stimuli and matching procedure
In the discovery sample and in the validation sample 1, the task was implemented in the software *Presentation* (https://www.neurobs.com/). Visual geometric stimuli were used as conditioned stimuli (RGB code: 142, 180, 227) on a black background. Heat pain stimulation was delivered with a thermode (Pathway System, CHEPS thermode, MEDOC, Israel, http://www.medoc-web.com, baseline temperature 35° and heating, cooling rise and fall rates of 70 and 40°/s) to the left volar forearm for 2.5 s. Individual temperature levels with an unpleasantness rating of VAS70 on a 0–100 VAS were determined using a calibration procedure and used subsequently (see Supplementary Methods: Calibration of aversive stimuli and ref. 78 for details). The maximum temperature of 50° was used as a safety limit to avoid potential tissue damage. An auditory stimulus was generated using the software *Audicity* software (v: 1.3.10-beta; http://www.audacity.sourceforge.net/) with the following parameters: sawtooth waveform profile, duration: 2.5 s, fade-in: 180 ms, fade-out: 300 ms. The volume was adjusted to match the unpleasantness of the pain stimulus in a separate calibration phase as follows. Immediately after the application of the heat stimulus, the auditory stimulus was delivered. Participants had to indicate whether the tone was less unpleasant, more unpleasant, or as unpleasant as the heat stimulus. Based on their response, the volume of the tone was automatically adjusted, but the procedure was stopped if both types of stimuli were equally unpleasant. This was repeated five times, and the mean volume level was used. As a final check after the matching procedure, an additional test was performed, in which each pain and tone stimulus was delivered three times and their unpleasantness was rated. If there was a considerable discrepancy,

the volume and/or the temperature were adjusted to ensure that stimuli were equally unpleasant.

In validation sample 2, similar geometric figures were used as CS. However, instead of the heat pain stimulation, visceral pain was used as US (see "Behavioral paradigm" above). The pressure-controlled rectal distension was elicited with a barostat system (ISOBAR3 device, G&J Electronics, Toronto, ON, Canada). The individually calibrated pressure for visceral pain was determined with a rating between 60 and 80 on a VAS100 scale (see Supplementary Material and in refs. 29,79,80). A similar system as in the other two studies (*Audicity* software) was used to elicit the auditory signal with the same parameters. A similar matching procedure ensured that the pain and unpleasant tone stimuli were matched in unpleasantness (see details above). The intensity of the unpleasant tone was adjusted until unpleasantness ratings of the two stimuli were comparable in two consecutive trials.

### Questionnaires
To assess individual psychological state and trait characteristics, participants filled different questionnaires regarding depression (ADS-K[81]), pain-related anxiety (Pain Anxiety Symptoms Scale – PASS-D[82]), and pain catastrophizing (Pain Catastrophizing Scale[83]). Additionally, participants provided fear ratings of pain and tone stimuli before the experiment on 0–100 VAS (verbal anchors: 0 = "not afraid at all" and 100 = "extremely afraid"). We calculated the difference between the two ratings, to further characterize individual differences.

### MRI preprocessing
All acquired resting-state data were preprocessed with our pre-registered, in-house built pipeline: the RPN-signature pipeline (version 0.2.6)[6]. The docker version of the pipeline can be found here (https://spisakt.github.io/RPN-signature/). The preprocessing utilized various tools from the *FSL*[84], *AFNI*[85], and *ANTs*[86] software libraries and employed *nipype*[87] to perform the following steps: brain extraction, boundary-based co-registration from functional to anatomical space, high precision co-registration from anatomical to standard space, motion correction, nuisance regression (24 motion regressors[88] and temporal SNR-based compcor[89] correction with 5 components), bandpass filtering (0.08–0.008 Hz) and scrubbing of high motion time frames (framewise displacement, FD > 0.15). We visually checked all the brain extraction and registration results. The brain masks were manually corrected in case of error. A functional brain atlas (data-driven parcellation derived with the BASC algorithm)[45] was used with 122 non-overlapping functional regions (the used atlas is available on the project GitHub page https://github.com/kincsesbalint/paintone_rsn/tree/master/data_in) and additionally, the global gray matter signal was also kept. The atlas regions were masked with gray matter and transformed back to the functional space. This approach is expected to decrease the heterogeneity introduced by registration across subjects and ensures that the resulting signal derives from gray matter voxels. An example region of interest of the amygdala is visualized in Supplementary Fig. 6. The individual connectivity matrices were calculated via partial correlation. We excluded participants based on our pre-registered criteria, namely, mean FD > 0.15 and percent of scrubbed volumes >25%. Based on these motion criteria, we excluded $n = 11$, $n = 7$, and $n = 5$ participants from the discovery sample, validation sample 1, and validation sample 2, respectively.

### Statistics and reproducibility
**Model discovery.** The timeseries of the regions were standardized and used to calculate individual partial correlation matrices resulting in 7503 functional connections. Our machine learning pipeline consisted of a k-best feature selection and a regularized linear model with L2 penalty (Ridge), executed in a cross-validation compliant manner similar to previous studies[6,7]. We used a nested leave-one-out cross-validation scheme for model training. The inner loop was used to optimize hyperparameters ($k$: number of selected features and alpha: regularization parameter) and the outer loop was to estimate model performance in the discovery phase. We fit a final model on the whole discovery sample with the identified best hyperparameters at the end of model training, to be tested in the external validation study. Predictive performance was estimated with the explained variance and the correlation between the observed and predicted values in the outer cv-loop. For significance testing, we used a nonparametric permutation test. Exact $p$ values were reported, $p < 0.05$ was considered as significant.

**External validation.** Following recommendations in predictive model development[41], we tested the model generalizability on the external validation sample 1 and validation sample 2 (external validity). The preprocessed connectivity matrices from validation sample 1 and validation sample 2 were fed into the developed model and model predictions were correlated with the observed DVC values. The association between the RCPL prediction and the observed values was estimated with correlation coefficients (one-tailed) for the two samples separately and for the pooled sample as well. Exact $p$ values were reported, $p < 0.05$ was considered as significant.

**Analysis of validators and confounder.** To test the effect of convergent, divergent validators, and confounders on model predictions, we evaluated conditional independence with the partial confounder test from the *mlconfound* software package[46]. Briefly, the correlation between the predicted value and the convergent, divergent validator or the confounder was tested given the correlation between the convergent, divergent validator or the confounder and the outcome variable. We tested against the null hypothesis that the model is not biased by the third variable (convergent validator, divergent validator, confounder). Convergent validators were contingency ratings after acquisition and extinction of $CS^+_{pain}$ and extinction learning. Divergent validators were tone-related learning (in acquisition and extinction), fear of pain, and pain anxiety. We investigated the following confounders: age, scores on depression questionnaires, and in-scanner motion (mean FD, max FD, median FD, percent of scrubbed volumes). Exact $p$ values were reported, $p < 0.05$ was considered as significant.

**Predictive importance.** To investigate the impact of brain connections and regions on the model performance, we used an externally validated leave-one-feature-out (LOFO) and a leave-one-region-out (LORO) approach. In detail, we excluded a single connection or all connections of a region (for LOFO and LORO, respectively) and retrained a model on the remaining subset of connections. Afterward, we compared the retrained model's explained variance to the original model's explained variance in the external validation samples. High values mean a drop in model performance estimation and indicate that the excluded connection/region has a high impact on the predictions.

### Reporting summary
Further information on research design is available in the Nature Portfolio Reporting Summary linked to this article.

## Data availability
Preprocessed fMRI timeseries data and behavior data are available on the project's GitHub page (https://github.com/kincsesbalint/paintone_rsn)[77]. The raw MRI data are deposited on the project OSF page (https://osf.io/b8znd/files/osfstorage).

## Code availability
All codes are available on the project's GitHub page (https://github.com/kincsesbalint/paintone_rsn)[77].

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

## Acknowledgements

The work is funded by the Deutsche Forschungsgemeinschaft (DFG, German Research Foundation) – Project-ID 422744262 – TRR 289 ("Treatment Expectation") (Gefördert durch die Deutsche Forschungsgemeinschaft (DFG) – Projektnummer 422744262 – TRR 289) and Project-ID 316803389 – TRR 1280 ("Extinction Learning").

## Author contributions

B.K., U.B., and T.S. conceived the study. Discovery study was planned and performed by K.F., K.S., and U.B.; validation study 1 was planned and performed by B.K., K.F., K.S., F.S., and U.B.; validation study 2 was planned and performed by R.J.P. and S.E. B.K. and T.S. conducted the analyses; B.K., K.W., U.B., and T.S. wrote the manuscript. K.S., D.T., R.J.P., F.S., S.E., K.W., B.K., U.B., and T.S. contributed to the interpretation, as well as manuscript revision. The whole study was supervised by B.K., U.B., and T.S.

## Funding

## Competing interests

The authors declare no competing interests.
