## [Peer Review File · Communications Biology]

Reviewers' comments:

Reviewer #1 (Remarks to the Author):

The study developed and validated a resting-state functional connectivity-based machine learning model of pain-related learning. The model showed good predictive performance in the discovery set and two independently collected validation sets, and seemed to be specific to pain learning rather than general aversive learning and not confounded by various potential confounders. The main contributors to the model were found to be the amygdala – parieto-occipital associative area, posterior insula – left sensorimotor cortex, angular gyrus - right anterior PFC and cerebellum (I-V) – posterior cingulum. My comments are as follows:

1. Based on the valence ratings, the CSs (including CS+) were not very unpleasant to participants. Before the acquisition phase, they were pleasant (< 50 on the valence rating scale); during the acquisition phase, they were a little unpleasant in the discovery set (slightly > 50), and at best neutral in the two validation sets (<= 50). These ratings suggest that the CSs are not “initially neutral geometrical figures”. They are either pleasant initially and never conditioned to be very unpleasant, or participants did not rate the valence according to the verbal anchors (50 means neutral). The conditioning task itself thus may not be a simple aversive learning task, but may involve other processes like rewarding. By the way, I recommend the authors report the ratings for CS+ and CS- in addition to the rating differences between CS+ and CS-.

2. The specificity of the model was assessed by partial correlation between the model predictions and aversive tone learning given pain learning in the discovery set. Will the results differ in the two validation sets? Also, the authors could test whether the model predictions still correlate with pain learning while controlling aversive tone learning. If not, the correlation between model predictions and pain learning would be driven by the correlation between pain learning and aversive tone learning.

3. Another approach to test the model specificity is to develop two models, one for pain-related learning, the other for aversive tone learning, and then examine whether the pain learning model can predict aversive tone learning (what the authors did in the paper) and whether the aversive tone learning model can predict pain-related learning.

4. Pain-related learning is related to the development and maintenance of chronic pain. The model in the paper predicts the magnitude of learned valence differences between CS+(pain) and CS-. Apart from this, learning rate, resistance to extinction and overgeneralization can also be related to pain chronification. What insights can this study provide to these aspects of pain learning? In the discovery and validation sets, the relationship between the magnitude of valence differences, learning rate, and resistance to extinction can be empirically tested. If they are indeed correlated, I think this study can have much more clinical implications.

5. The authors should provide more information about the behavioral task like the stimulus parameters for US. Simply referring to other sources like the preregistration and another paper is

not reader-friendly. The authors should report the hyperparameters of the final model, including the k in the best k feature selection, and the α in the ridge regression. The k can be inferred from the Neurobiological plausibility section, but the α was not reported in the manuscript.

6. A section on the limitations of the study is needed.

Reviewer #2 (Remarks to the Author):

Kinces et al. use resting state to derive a pain-related learning predictor. This work benefits from preregistration and code sharing, which help to address some concerns about overfitting. This is especially important as brain-behavior relationships generally appear to be quite unstable unless using very large sample sizes as outlined by Scott Marek's paper in Nature. I list concerns below.

1) Because the order is introduction, results, discussion, then methods, you cannot assume that people will first read the methods section before results. Jumping into the results section, you don't give much information about the conditioning paradigm. As a result, it was challenging for me to follow along with aspects of the results section. It might be beneficial to swap the infographic in Fig 1A with an infographic for the experimental design. The infographic used in the preregistration document (Fig1 preregistration) might work nicely.

2) Related to the previous comment, much more information needs to be included in the manuscript, particularly the methods. It is possible I am just missing something, but what machine was used to create pain, where were people stimulated on the body, what temperatures were used and how did you decide on those temperatures?

3) I potentially missed this in the methods, but how were heat pain stimuli and auditory stimuli individually calibrated? I see from the methods that they were, but I would like additional information. It appears from supplemental table 2 that the unpleasantness for pain was consistently higher than for tone. Additionally, the fear of the US is different for pain and tone. For some of the analyses, the use of contingency learning for aversive tone is used as a control, and can be used to argue for specificity of RCPL. However, since pain is more unpleasant and more fear evoking than tone, is aversive tone actually a good control? Ideally, the tone would be equally aversive and salient as the painful stimuli.

4) I am somewhat confused as to the goal of RCPL. Based on the name, the 'resting-state functional connectivity signature of pain-related learning' should be specific for pain learning, or else it is an index of general conditioning or more specifically fear conditioning. If I understand correctly, tone-related threat learning is treated as a positive validator. To me, an association of pain related learning with aversive tone related learning doesn't suggest 'generalizability' but rather a lack of specificity.

5) RCPL is related to aversive tone learning. You then say that this was driven by association between pain-related and aversive tone related learning. Then you say "predictions may primarily reflect pain-specific learning rather than aversive learning in general". This is what I would call cautious language ('may primarily reflect'), which is contrasted to other parts of the paper (abstract: "proposed model is specific for pain"; Results sub-heading "The signature was specific to pain-related learning"; Discussion: "we found no evidence for generalization to aversive tone-

related learning”). It seems a lot more caution is warranted with claims of specificity. Additionally, it is unfortunate, but understandable that the methods used for mlconfound do not appear to be currently published. This is unfortunate that such an important part of the methods for this manuscript relies on a method that hasn’t yet finished peer review. However, I understand completely that reviews have been taking a long time over the past few years. Because some of the methods haven’t yet been published, I would recommend spending much more time in the manuscript going over the approach, since the average reader will not be familiar. Additionally, I would challenge the authors about how specific a signature for pain related learning can be if it is correlated to non-pain learning. This has important implications for the neuroscience, because it is unclear if the brain association is for the pain-specific parts of the signature, or the general aversive learning elements of the signature.

6) It is excellent that you fix model weights and preregister. Looking at the correlation values between samples shows that performance in discovery model for RCPL is much better than in external samples (Rsquared difference 51.84 vs .11). The preregistration shows that elements of the models prediction are minimally accurate, but the performance drop off is quite alarming and suggests a non-trivial amount of overfitting in the model. At the end of the day, you all do much more than most to address overfitting by showing model performance after preregistration. I think the discussion would benefit from explicitly talking about why model performance drops off so much, how different models might perform etc.

7) You should report the mean squared error (or root) for predictive performance. See PMID: 31774490 Russ Poldrack paper for why this is superior to showing correlation between predicted and observed values. I especially want to see mean squared error because the predicted and observed DVC shown in figure 1 seem to be quite different. The first tell of this is that you had to use different scales for the x and y axis in Fig1C. I am curious to see how close the model gets to actually predicting the observed DVC values. Importantly, Poldrack argues that variance explained is not appropriate for out-of-sample testing.

8) Figure 3A. What do the colors next to certain areas refer to? Are these networks? You should have some look up table or guide for these networks. The purple network looks somewhat alarming (if it is a network). It features a lot of areas with notoriously low SNR.

9) The amygdala is notoriously challenging to localize with functional imaging. Based on the images provided, and the data driven parcellation used, I am somewhat skeptical how localized an ROI is to the amygdala. More anatomical data should be displayed to prove that you are actually looking at the amygdala. Can you also please confirm the amygdala peak (you report 3.06 -7.47 -19.9). I see you note that the center of mass can lie outside of the mass for bilateral parcels, in such cases, reporting the center mass is meaningless and somewhat confusing as it does not describe the ROI (maybe report left and right center mass). It might be available on your github, but would you be able to point to a .nii file containing the ROIs you used? You should provide additional information to demonstrate you are actually looking at the region you claim you are.

10) You should be defining the networks you are using in the discussion. What do you mean by parietal CEN?

11) Were the same M=122 regions used for all datasets? What dataset was used to generate the regions? How much would these regions change if they are estimated in a different dataset? Just curious, why did you choose to make your own parcellation instead of relying on a published parcellation?

12) Figure 3B. If you want to get a sense of dynamic range then you should pick a different color scaling. The difference between 10 and 5.2 is way too small to highlight dynamic range. If you want to just show all the areas that were significant, then just binarize the map.

13) In the outlook section you talk about applying the RCPL to diverse populations. Can you give an example of how this would be applied to these populations and how this would be better than using behavioral data alone to calculate pain DVC? I would like to know how you see a signature like this being used.

14) You say the cerebellum is known to be involved in fear condition behavior and that the cerebellum is a promising new target. This is bizarre to say. It is similar to saying, the cerebral cortex is known to be involved in pain and would be a promising target for research. Yes this is true, but it lacks any specificity. The cerebellum is enormous and has a similar surface area to the entire cerebral cortex.

15) I think the writing, especially in the introduction, does not do a good job of setting up a research question.

16) The text for 'preregistered' used in some infographics is pixelated.

17) SFig1 has an odd partially cut off '2' on the bottom left.

Reviewer #3 (Remarks to the Author):

In this study, Kincses et al. developed and evaluated a resting state functional brain connectivity based predictive marker of individual differences in pain-related learning, as measured by an established classical conditioning paradigm. For this they used 8 - 10 mins of resting state data collected prior to an established conditioning task to predict learning performance on the task, quantified as differential valence ratings (i.e., learning-induced increase in pain ratings; and sound ratings as a control condition for this study). They trained a machine learning model on whole brain data between 122 functionally defined regions to predict valence ratings reflecting pain related learning. The ML model was developed and tuned in the discovery sample (n = 25), then locked and preregistered, and then tested in 2 independent samples (n = 26 and n = 23) performing the equivalent conditioning paradigm.

The model's predictive accuracy seems to be on par with several other efforts in genetic or brain marker development of predicting between-subject traits. The study results have a clear clinical relevance as a potential easily applied prognostic and diagnostic marker (requiring only resting state fMRI data) and neuroscientific merit by advancing our understanding of brain mechanisms of pain threat learning. The study implements several guardrails against ML-based issues (overfitting, effect of confounds, misinterpretation), which is commendable.

I have several reservations, as listed in the following.

Given the massive translational clinical relevance of this signature – as also emphasized by the authors in their introduction and discussion, as well as the ease of application to new datasets (“As the RCPL signature is derived solely from resting-state functional connectivity, and therefore

requires only a 10-minute non-invasive scan without the use of noxious stimuli or time-consuming learning paradigms, it is a highly accessible tool for these clinical and translational purposes”) it seems like a missed opportunity not to have tested this signature in a publicly (or otherwise) available sample of people with chronic pain. Such a test – even if only as a preliminary validation in a relevant clinical sample - would likely strengthen the claims of the paper.

Re. specificity of the signature to predict pain vs. sound: I might not be understanding the Generalization test, but could not an alternative plausible explanation be that transfer learning (pain to sound) is a potential mechanism involved in aversive learning and therefore the results support a more general aversive learning? Some additional clarification as to why the presented result strongly favors pain-specific interpretation might be needed.

Given how similar the conditioned ratings for pain and sound are (suppl. Fig 1) would it make sense to consider training on the sound data and examine whether a sound learning signature is specific to sound or also sensitive to pain to further isolate the pain-specificity of the developed signature?

Re. number of subjects used in the paper, it is a bit confusing to keep seeing the size of the initial sample (before exclusions) reported ($n = 99$; for example in behavioral results table 1, and in Results first paragraph) when actually $n = 74$ were used in signature development and validation (after exclusions) across the 3 sub-samples (model development and internal cross-validation, external validations: $n = 25$; $n = 26$, $n = 23$) and for observed-predicted performance calculation and the corresponding scatter plots (Figure 1, Suppl. Figure 1). It might be more accurate /less confusing to primarily report $n = 74$ (and its sub-sample numbers).

Point-by-point response to reviewers' comments

Reviewer #1

The study developed and validated a resting-state functional connectivity-based machine learning model of pain-related learning. The model showed good predictive performance in the discovery set and two independently collected validation sets, and seemed to be specific to pain learning rather than general aversive learning and not confounded by various potential confounders. The main contributors to the model were found to be the amygdala – parieto-occipital associative area, posterior insula – left sensorimotor cortex, angular gyrus - right anterior PFC and cerebellum (I-V) – posterior cingulum. My comments are as follows:

Rev 1/Q1

1. Based on the valence ratings, the CSs (including CS+) were not very unpleasant to participants. Before the acquisition phase, they were pleasant (< 50 on the valence rating scale); during the acquisition phase, they were a little unpleasant in the discovery set (slightly > 50), and at best neutral in the two validation sets (<= 50). These ratings suggest that the CSs are not “initially neutral geometrical figures”. They are either pleasant initially and never conditioned to be very unpleasant, or participants did not rate the valence according to the verbal anchors (50 means neutral). The conditioning task itself thus may not be a simple aversive learning task, but may involve other processes like rewarding. By the way, I recommend the authors report the ratings for CS+ and CS- in addition to the rating differences between CS+ and CS-.

We are grateful to the Reviewer for highlighting the good predictive performance of our model and the potential of a bias in baseline valence ratings, which was the key motivation for our use of differential valence ratings as the prediction target. Such an initial bias, and more generally a high inter-individual variance in ratings, are commonly observed (see Lonsdorf & Merz, 2017 for discussion), including in the aversive learning paradigm we used, which employs simple, randomized geometric figures as CSs, to minimize subjective associations at baseline. However, our study did not focus on the subjective experience of (un)pleasantness, per se, but on its change as a result of learning the association between the CS and the aversive (unconditioned) stimuli. The statistically significant within-participant change in valence is comparable to those reported in the literature (Koenen et al., 2018; Schlitt et al., 2021; Zlomuzica et al., 2015) and clearly indicates successful, statistically significant aversive learning. Exactly this change is what our target variable (differential valence) is based on, when training our brain-based model. Furthermore, our model successfully generalized to validation sample 2, which used a scale that anchored 0 to “neutral” valence, suggesting that the predictions are not influenced by the reporting of CS valence. As requested, we provide additional tables about the ratings for each CS type in each phase as *Supplementary Table 5*.

Rev 1/Q2

2. The specificity of the model was assessed by partial correlation between the model predictions and aversive tone learning given pain learning in the discovery set. Will the results differ in the two validation sets? Also, the authors could test whether the model predictions still correlate with pain learning while controlling aversive tone learning. If not, the correlation between model predictions and pain learning would be driven by the correlation between pain learning and aversive tone learning.

Following the Reviewer's suggestion, we have extended our specificity analysis to the external validation samples (*Supplementary Material: Model validators in the validation sample*). The analysis of the validation samples corroborated our initial results; we found no evidence for conditional dependency between the predictions and aversive tone learning ($p_{VS1}=0.32$, $p_{VS2}=0.82$, $p_{VS1+VS2}=0.35$, see details in *Supplementary Table 6*). Note, however, that we did not use partial correlation, as corresponding p-values are known to be unstable and highly sensitive to even the slightest violation of

normality and linearity. Despite early warnings of this problem (Korn, 1984), the magnitude of the problem may not be fully appreciated. To this end, we tested the statistical conditional independence of the predictions from aversive tone learning (conditioned on pain learning) with a dedicated, peer-reviewed statistical test (Spisak, 2022) (see Rev1/Q4 and Rev 2/Q4-5, Rev3/Q2 for more details), which requires minimal assumptions.

Rev 1/Q3

3. Another approach to test the model specificity is to develop two models, one for pain-related learning, the other for aversive tone learning, and then examine whether the pain learning model can predict aversive tone learning (what the authors did in the paper) and whether the aversive tone learning model can predict pain-related learning.

We agree that comparing models trained separately for pain and tone could provide additional insight into shared and distinct mechanisms. However, we found that the discovery sample was not sufficiently powered to achieve significant predictive performance for aversive tone learning with nested cross-validation (Supplementary Material *Model training on tone related learning as target*). Such possibly false negative training results are among the known, inherent limitations of small sample model discovery in machine learning (Spisak et al., 2023; Varoquaux, 2018). While our pre-registered external validation is well positioned to confirm our true positive discoveries (predictive model of pain-related learning), it is not able to confirm whether the lack of significant predictions for tone-related learning is a false negative observation (Gallitto et al., 2023). In other words, while the present study does not allow for the analysis recommended by the Reviewer, it has laid the groundwork for scaling up the model evaluation process to sample sizes that are sufficient for addressing questions of shared and specific mechanisms. Limitations for further analysis due to the small discovery sample are now discussed in more detail on page 19 (see also our response to Rev 1/Q6).

Rev 1/Q4

4. Pain-related learning is related to the development and maintenance of chronic pain. The model in the paper predicts the magnitude of learned valence differences between CS+(pain) and CS-. Apart from this, learning rate, resistance to extinction and overgeneralization can also be related to pain chronification. What insights can this study provide to these aspects of pain learning? In the discovery and validation sets, the relationship between the magnitude of valence differences, learning rate, and resistance to extinction can be empirically tested. If they are indeed correlated, I think this study can have much more clinical implications.

We agree that disentangling the ‘magnitude’ of learning, learning rate, extinction, and generalization in terms of their neural signatures in resting state connectivity is an important step toward establishing a more comprehensive view of chronic pain. As suggested, we assessed individual learning rates (by fitting Rescorla-Wagner models, see Supplementary Material: *Assessment of learning*) and resistance to extinction. Both variables are highly correlated with differential valence rating ($R^2=46.1\%$ and $R^2=84.6\%$ for extinction and learning rate, respectively). Statistical tests of conditional independence did not provide evidence that the model generalizes to these measures (extinction: $p_{DS}=0.86$, $p_{VS1}=0.54$, $p_{VS2}=0.62$, $p_{VS1+VS2}=0.33$, learning rate: $p_{DS} = 0.5$). However, it should be noted that with high association between the prediction target and the third variable (i.e., learning rate or extinction), the power of the applied conditional independence test drops drastically (see *Supplementary Figure 3*), meaning that deciding whether the proposed model generalizes to ‘magnitude’ of learning, learning rate and resistance to extinction is only possible with larger samples. We now discuss this as a limitation in the revised manuscript (see Rev 1/Q6).

Supplementary Figure 3. Power calculation for specificity analysis.

Rev 1/Q5

5. The authors should provide more information about the behavioral task like the stimulus parameters for US. Simply referring to other sources like the preregistration and another paper is not reader-friendly. The authors should report the hyperparameters of the final model, including the k in the best k feature selection, and the α in the ridge regression. The k can be inferred from the Neurobiological plausibility section, but the α was not reported in the manuscript.

We agree with the Reviewer and have now updated the main text and included additional information on the paradigm, stimuli, and their calibration and the matching procedure, as well as the final model parameters in the Methods section and in the Supplementary Material as shown below:

Page 7, line 10-11

“The final regularized linear model (Ridge) was fitted on the discovery sample and contained ten predictive connections with a regularization parameter of $\alpha=0.001$.”

Page 23, line 26- page 24, line 26:

“Stimuli and matching procedure

In the discovery sample and in the validation sample 1, the task was implemented in the software Presentation (<https://www.neurobs.com/>). Visual geometric stimuli were used as conditioned stimuli (RGB code: 142, 180, 227) on a black background. Heat pain stimulation was delivered with a thermode (Pathway System, CHEPS thermode, MEDOC, Israel, <http://www.medoc-web.com>, baseline temperature 35° and heating, cooling rise and fall rates of 70 and 40°/s) to the left volar forearm for 2.5 seconds. Individual temperature levels with an unpleasantness rating of VAS70 on a 0-100 visual analogue scale (VAS) were determined using a calibration procedure and used subsequently (see Supplementary Material and Forkmann et al.⁷⁷ for details). The maximum temperature of 50° was used as a safety limit to avoid potential tissue damage. An auditory stimulus was generated using the software Audacity software (v: 1.3.10-beta; <http://www.audacity.sourceforge.net/>) with the following parameters: sawtooth waveform profile, duration: 2.5s, fade-in: 180ms, fade-out 300ms. The volume was adjusted to match the unpleasantness of the pain stimulus as follows. Immediately after the application of the heat stimulus, the auditory stimulus was delivered. Participants had to indicate whether the tone was less unpleasant, more unpleasant, or as unpleasant as the heat stimulus. Based on their response, the volume of the tone was automatically adjusted, but the procedure was stopped if both types of stimuli were equally unpleasant. This was repeated five times, and the mean volume level was used. As a final

check after the matching procedure, an additional test was performed, in which each pain and tone stimulus was delivered three times and their unpleasantness was rated. If there was a considerable discrepancy, the volume and/or the temperature were adjusted to ensure that stimuli were equally unpleasant.

In the validation sample 2, similar geometric figures were used as CS. However, instead of the heat pain stimulation, visceral pain was used as US (see Behavioral paradigm above). The pressure controlled rectal distension was elicited with a barostat system (ISOBAR3 device, G&J Electronics, Toronto, ON, Canada). The individually calibrated pressure for visceral pain was determined with a rating between 60 and 80 on a VAS100 scale (see Supplementary Material and in Koenen et al. and Pawlik et al.^{29,79}). A similar system as in the other two studies (Audicity software) was used to elicit the auditory signal with the same parameters. A similar matching procedure ensured that the pain and unpleasant tone stimuli were matched in unpleasantness (see details above). The intensity of the unpleasant tone was adjusted until unpleasantness ratings of the two stimuli were comparable in two consecutive trials.”

Supplementary Material Page 2, line 1-23:

“Calibration of aversive stimuli

In the first part of the calibration, individual thresholds for heat pain and the unpleasant tone were determined with the method of limits. For heat pain, the temperature was slowly increased (1°C/s) starting at a baseline of 35°C and the participants had to indicate when the sensation had become painful for the first time by pressing a button. The upper limit for heat stimuli was set to 50°C. A similar method was used to calibrate the auditory stimuli. Participant had to indicate when the gradually increasing loudness became unpleasantly loud. A maximum loudness of 120dB was never exceeded. Both procedures were repeated three times and the average was used as heat pain and unpleasant tone threshold. The participants then received ten stimuli of different temperature levels two times (ranging from (pain threshold - 1°C) to (pain threshold + 3.5°C) with 0.5°C degree increments) resulting in 20 stimuli in total. After each stimulus, unpleasantness was rated. A linear regression model was fitted to the data to determine the temperature level corresponding to VAS70. This level was subsequently used in the matching procedure.

For visceral pain, different pressure levels (increments of 5 mmHg) with a duration of 30s were used to determine the visceral pain threshold. Participants provided the perception of the stimulus on a scale from 1 to 4 (1 = no perception over 2 = likely perception and 3 = urge to defecate to 4 = painful perception). The pressure was increased until the visceral pain threshold was reached (rating of 4). The upper pressure limit was set to 55 mmHg. The procedure was terminated when the participant reported the first sensation of pain. During the calibration, rectal distension stimuli delivered in 5 mmHg increments and rated until the predefined level (between VAS60-80) was reached. The starting pressure was 5 mmHg below the visceral pain threshold.”

Rev 1/Q6

6. A section on the limitations of the study is needed.

Following the Reviewer’s suggestion, we have added the following limitation section:

Page 19, lines 10-28:

“Limitations

Low reliability of self-reported outcome measures can be a significant limitation in terms of the power and reliability of brain-based predictive models^{75,76}. This may also apply to the model presented here, which was trained to predict pain-related learning indexed as differential valence ratings, i.e., based on subjective self-reports. Thus, the reported predictive performance is likely to be lower than what could optimally be achieved with more reliable target measures.

Our study used a small discovery sample combined with pre-registered external validation. While this approach allows for an efficient and reliable construction of brain-based predictive models³⁷, our power

calculation (Supplementary Figure 3) indicated that moderate to large samples⁴⁴ will be needed to thoroughly characterize the model in terms of its specificity to pain-related mechanisms, its relationship to other aspects of pain-related learning (e.g., resistance to extinction), and its validity in clinical populations.

Disentangling the common and distinct mechanisms of pain- and tone-related learning is further limited in the present study by their potential interactions in our paradigm, through transfer learning and cross-modality generalization.

Like any fMRI-based study, our study is also limited by low SNR in regions prone to artifacts (e.g. susceptibility artifacts), including regions that are known to be involved in pain-related learning, such as the orbitofrontal cortex. Results relating to these regions should therefore be interpreted with caution.”

Reviewer #2

Kinces et al. use resting state to derive a pain-related learning predictor. This work benefits from preregistration and code sharing, which help to address some concerns about overfitting. This is especially important as brain-behavior relationships generally appear to be quite unstable unless using very large sample sizes as outlined by Scott Marek's paper in Nature. I list concerns below.

We would like to thank the Reviewer for highlighting the importance of pre-registered external validation and code sharing for brain-based predictive models. We are convinced that implementing these standards is the key for producing brain-behavior models that are robust and reliable enough for translational research applications.

Rev 2/Q1

1) Because the order is introduction, results, discussion, then methods, you cannot assume that people will first read the methods section before results. Jumping into the results section, you don't give much information about the conditioning paradigm. As a result, it was challenging for me to follow along with aspects of the results section. It might be beneficial to swap the infographic in Fig 1A with an infographic for the experimental design. The infographic used in the preregistration document (Fig1 preregistration) might work nicely.

We agree with the Reviewer and have added information on the conditioning paradigm to the results section, including a reference to the requested infographic:

Page 5, line 6-11:

"In brief, the unconditioned stimuli (US pain and tone) were preceded with geometric figures as conditioned stimulus (CS) and participants rated the unpleasantness of these conditioned stimuli (via valence ratings). Another geometric figure that was never followed by a US served as control (CS⁻). Over the course of the experiment, participants learned to associate the CSs with the USs which was reflected in the change of valence ratings (see Methods and Figure 1 for more detail)."

Page 5, line 14-18:

"Individual pain-related learning performance was characterized by differential valence ratings (see Equation 1), i.e., the degree to which the valence difference between CS^{+pain} and CS⁻ (see Equation 1) changed over the course of the experiment for a given individual. Differential valence ratings served as the target measure for the discovery and validation of our brain-based predictive model of pain-related learning."

Rev 2/Q2

2) Related to the previous comment, much more information needs to be included in the manuscript, particularly the methods. It is possible I am just missing something, but what machine was used to create pain, where were people stimulated on the body, what temperatures were used and how did you decide on those temperatures?

We apologize for the lack of detail about the procedures. In the discovery sample and validation sample 1, a contact thermal stimulator was used (Pathway System, CHEPS thermode, MEDOC, Israel) on the left volar forearm. In the validation sample 2, visceral pain was induced by a barostat system (pressure controlled rectal distension; ISOBAR3 device, G&J Electronics, Toronto, ON, Canada). The used pain stimulus intensities were individually calibrated after the resting state scan but before the main experiment and the aversive tone stimuli were matched in unpleasantness to the painful thermal stimulation in a separate procedure. The maximum temperature of 50°C was used as a safety limit to avoid tissue damage. The reason we omitted some of the details in the original manuscript was that they were already described in detail in a previous paper (Forkmann et al., 2023). However, we fully agree

with the Reviewers' comment (see also Rev 1/Q5) that omitting these details detracts from readability. We now provide additional information in the methods part, and in the Supplementary Material (Methods: *Stimuli and matching procedure* section and Supplementary Material Methods part: *Calibration of aversive stimuli*).

Rev 2/Q3

3) I potentially missed this in the methods, but how were heat pain stimuli and auditory stimuli individually calibrated? I see from the methods that they were, but I would like additional information. It appears from supplemental table 2 that the unpleasantness for pain was consistently higher than for tone. Additionally, the fear of the US is different for pain and tone. For some of the analyses, the use of contingency learning for aversive tone is used as a control, and can be used to argue for specificity of RCPL. However, since pain is more unpleasant and more fear evoking than tone, is aversive tone actually a good control? Ideally, the tone would be equally aversive and salient as the painful stimuli.

We agree that the information on these aspects of the study should have been more detailed. In response, we now provide additional information on the calibration and matching procedure together with the details requested in the previous comment (see R1/Q5, Methods: *Stimuli and matching procedure* and Supplementary Material Methods part: *Calibration of aversive stimuli*). Regarding the reported unpleasantness rating, the supplementary table includes the values of the habituation phase (in discovery sample and validation sample 1). The mean value for pain and tone do appear to be different, but this difference is only statistically significant in validation sample 1 (Kolmogorov-Smirnov test: D(25)=0.32, p=0.16; D(26)=0.5, p=0.003; D(23)=0.35, p=0.12 for the discovery sample, validation sample 1 and validation sample 2 respectively). Furthermore, the two stimulus unpleasantness scores were comparable during the acquisition (Kolmogorov-Smirnov test: D(25)=0.24, p=0.48; D(26)=0.15, p=0.93; see the part of the Supplementary Table 2 below for the discovery sample and validation sample 1, respectively). We have now added the unpleasantness ratings in the acquisition phase to the *Supplementary Table 2*. In addition, fear of the different aversive stimuli was measured before the actual experiment (see *Supplementary Figure 1*). Based on our confounder analysis, the effect of fear of the different stimuli does not seem to have an effect on the prediction.

The parts in red of the table were added to Supplementary Table 2:

US unpleasantness	habituation	pain	59 (28)	69 (15)	-
		tone	50 (19)	54 (15)	-
	acquisition	pain	54 (16)	52 (16)	74 (22) ⁺⁺
		tone	51 (17)	51 (16)	62 (23) ⁺⁺

Part of the Supplementary Table 2

Rev 2/Q4

4) I am somewhat confused as to the goal of RCPL. Based on the name, the 'resting-state functional connectivity signature of pain-related learning' should be specific for pain learning, or else it is an index of general conditioning or more specifically fear conditioning. If I understand correctly, tone-related threat learning is treated as a positive validator. To me, an association of pain related learning with aversive tone related learning doesn't suggest 'generalizability' but rather a lack of specificity.

We would like to clarify that the tone-related learning scores (differential valence ratings) served as a negative validator and apologize that this was not made clearer in the original submission. We now explicitly state this at multiple points in the manuscript. For instance, we have modified the headings on *Figure 2*, where the use of the term "Generalization" was indeed misleading, falsely suggesting that all

variables tested with the “partial generalization test” were considered positive/convergent validators (Spisak, 2022); see also our response to Rev 1/Q2 and the reply to the Editor’s first comment). Furthermore, as there seems to be no consensus in the community on the meaning of specificity and generalizability of machine learning models, we now clearly state our definition of these terms in the revised manuscript. We thoroughly updated the results section on specificity, updated *Figure 2* and provide an additional table, *Table 2*:

Page 8 line 17- page 9 line 32

“Specificity to pain-related learning and confounding bias

To be considered as a valuable tool for basic and translational clinical research, our findings on the externally validated, out-of-sample predictive performance of our proposed model must be complemented by an initial investigation of the model’s convergent and divergent validity⁴¹. Here we aimed to characterize the model’s (i) generalizability to other measures of pain-related learning (convergent validity to contingency ratings and extinction learning), (ii) its specificity to pain (divergent validity to tone-related learning) and (iii) confounding bias (divergent validity to various confounding variables, such as fear of pain or in-scanner motion artifacts).

Our criteria for specificity, generalizability, and confounding go beyond the common practice of drawing conclusions simply from bivariate associations between the model predictions and the (convergent and divergent) validator variables. Instead of this arguably misleading approach, we focus on the conditional independence structure across all three variables involved (i.e., target, prediction, validator) with a dedicated statistical test⁴⁶. In our definition, predictions are specific to the target variable, if they are conditionally independent of the validator variable, i.e., their (marginal) association is not stronger than would be expected from the association between the target and the validator variable. Conversely, if the predictions are conditionally dependent on the validator variable (i.e., the validator explains variance in the predictions over and beyond the expected marginal association), we say that the predictions generalize to the validator variable. In our view, confounding bias is equivalent to an (unwanted) generalization to a divergent validator (for more details on our approach see Spisak et al.⁴⁶). Differences between specificity assessment with bivariate measures and conditional independence can be illustrated by testing the convergent validity of our model with another measure of pain-related learning such as contingency rating. In contrast to the affective aspect of learning (measured by the change in valence ratings, Equation 1), contingency captures the cognitive component of pain-related learning. While the relationship between contingency learning and the prediction of our model was significant ($R^2 = 32\%$, $p=0.003$), we also have to take into account that contingency learning and valence learning were also highly correlated ($R^2=17\%$, $p=0.038$). Therefore, to demonstrate that the model generalizes to contingency learning, we need to reject the null hypothesis of conditional independence between the predictions and contingency learning (given the association of both with differential valence rating). The specific statistical test was able to reject this null hypothesis with $p=0.05$, meaning that the association between the prediction and contingency learning was not only significantly different from zero, but it was significantly higher than expected from the associations between valence learning and contingency learning (Figure 2 Convergent validity, Table 2).

Similarly, the predictions of the RCPL signature were significantly associated with differential valence ratings (internally validated estimate: $R^2=51\%$, $p<0.001$, nested LOPO cross-validation) not only during the acquisition phase, but also during extinction ($R^2=18.6\%$, $p=0.03$) (i.e., the recovery of valence ratings after cessation of conditioning (CS⁺-US pairing)). However, when testing for conditional independence between extinction and the model predictions, we found no evidence that this association was more than a consequence of the correlation between acquisition and extinction learning alone ($R^2=46\%$, ($p=0.86$; Figure 2 Convergent validity, Table 2).

The model predictions were also significantly correlated with valence learning induced by the aversive tone ($R^2=18.9\%$, $p=0.03$). However, the specific statistical test suggested that this association could be explained as a secondary consequence of the association between pain-related learning and learning induced by the aversive tone ($R^2=49\%$). Thus, our analysis could not provide evidence against the specificity of our model for pain-related learning. Note that if our model would primarily capturing

general aversive learning (i.e., neural mechanisms shared between pain and tone learning), its predictions would also be in conditional dependence with tone learning (Figure 2 Divergent validity, Table 2).”

Figure 2. Analysis of convergent validity, divergent validity, and confounder bias.

Convergent validity: Testing the generalizability of the model to the cognitive aspect of learning suggests that the RCPL signature captures pain-specific learning, as reflected in the rejection of conditional independence of contingency ratings ($p=0.05$). However, model prediction has not generalized to extinction learning ($p>0.05$).

Divergent validity: Model prediction was tested against tone related aversive learning. The model was not driven by tone learning, suggesting that the model primarily captures pain-related learning and not tone. Other divergent validators were also tested, including fear of pain/tone, pain catastrophizing, and pain anxiety, but none of them significantly biased the model.

Confounding bias: None of the investigated confounder variables including age, in-scanner motion parameters (mean, median and maximum framewise displacement, percent of scrubbed volumes), and depression score biased our model (see *Supplementary Material 1*).

The thickness and the opacity of the lines are proportional to the R^2 values. Percentages refer to the coefficient of determination R^2 between the two variables. $p < 0.05$ indicates that the model prediction is significantly associated with the variable over and above the ‘baseline’ association between the observed pain-related differential valence ratings (DVC, Equation 1) and the variable itself (e.g., contingency learning). The observed and predicted valence learnings were significantly correlated $R^2=51\%$, $p<0.001$.

Abbreviations: Conting. Learning – CS^+_{pain} contingency after acquisition training, Extinction learning – differential valence rating in extinction learning (see prereg for more detail), Tone learning– differential valence rating for tone (see Equation 1), Fear –fear of pain (see Questionnaires), Motion – head motion during the resting-state fMRI (mean framewise displacement (FD)).

		R ² with pain learning (%)	R ² with model prediction (%)	Significance of partial generalization test (P)	
Convergent validity	Contingency CS ⁺ _{pain,acq}	17.4	31.9	0.05*	
	Contingency CS ⁺ _{pain,ext}	1.4	25.3	0.03*	
	Extinction learning _{pain}	46.1	18.6	0.86	
Divergent validity	Tone learning	49.1	18.9	0.80	
	Extinction learning _{tone}	33.5	7.5	0.92	
	Fear of pain	2.9	6.7	0.30	
	Fear of tone	0.5	0.0	0.96	
	Pain catastrophizing	0.0	0.0	0.91	
	Pain anxiety	PASS D1	0.3	0.0	0.99
		PASS D2	6.5	0.8	0.78
		PASS D3	2.0	1.1	0.64
PASS D4		0.4	1.1	0.62	
Confounder bias	Age	1.3	3.5	0.41	
	Motion	Mean FD	0	2.0	0.49
		Median FD	0.2	1.8	0.52
		Max FD	2.5	7.5	0.24
		Percent scrubbed	0.2	1.0	0.63
	Depression	0.5	0.7	0.69	

Table 2 Result of partial generalization test.

We tested convergent and divergent validators, and confounder variables for model prediction. See main text for detailed description of convergent, divergent validators, and confounder bias.

Rev 2/Q5

5) RCPL is related to aversive tone learning. You then say that this was driven by association between pain-related and aversive tone related learning. Then you say “predictions may primarily reflect pain-specific learning rather than aversive learning in general”. This is what I would call cautious language (‘may primarily reflect’), which is contrasted to other parts of the paper (abstract: “proposed model is specific for pain”; Results sub-heading “The signature was specific to pain-related learning”; Discussion: “we found no evidence for generalization to aversive tone-related learning”). It seems a lot more caution is warranted with claims of specificity. Additionally, it is unfortunate, but understandable that the methods used for mlconfound do not appear to be currently published. This is unfortunate that such an important part of the methods for this manuscript relies on a method that hasn’t yet finished peer review. However, I understand completely that reviews have been taking a long time over the past few years. Because some of the methods haven’t yet been published, I would recommend spending much more time in the manuscript going over the approach, since the average reader will not be familiar. Additionally, I would challenge the authors about how specific a signature for pain related learning can be if it is correlated to non-pain learning. This has important implications for the neuroscience, because it is unclear if the brain association is for the pain-specific parts of the signature, or the general aversive learning elements of the signature.

We would like to thank the Reviewer for pointing out the importance of assessing model specificity, both for our work and for neuroscience research in general. Our method ‘*mlconfound*’ addresses this issue with mathematical rigor and it indeed serves as a cornerstone of the present paper. It is therefore very unfortunate that we mistakenly cited the preprint of the work, even though the paper underwent open peer-review by leading experts in the field (see review #1 and #2) and has been published in 2022 in the journal GigaScience. We apologize for the confusion caused by our mistake.

As the community has not yet agreed on precise definitions for specificity, generalizability, and confounding bias in the case of brain-based predictive models, we understand that clear communication of both methods and results are crucial. We have now clarified our definitions (see our answer to the previous remark) and added more details on the methodology (Methods: *Stimuli and matching procedure* section and Supplementary Material Methods part: *Calibration of aversive stimuli*). Furthermore, based on your comment, we have now extended our analyses with power calculations (*Supplementary Figure 3* based on (Spisak, 2022)) which suggest that with our sample sizes, we are only well powered (>80%) to reliably detect the lack of specificity if the prediction explains a large amount of variance (40%) in tone-related learning. This means that our results on specificity need to be interpreted with caution, and that larger samples are needed to disentangle pain-specific parts of the signature from those related to general aversive learning. Nevertheless, we believe that our results on specificity are promising enough to motivate the community to “scale up” our study and test specificity, generalizability, fairness, and ultimately clinical validity with sufficient power. We have adjusted the wording throughout the manuscript accordingly and toned down claims of specificity where needed. Please see also our reply to the previous comment and below the most important examples:

Abstract:

“The pre-registered external validation indicates that the proposed model explains 8-12% of the inter-individual variance in pain-related learning.”

Discussion:

Page 17, line 4-7:

“While our power analysis (Supplementary Figure 3) suggests that larger samples are needed to reliably disentangle general and pain-specific mechanisms in aversive learning, our results provide a promising basis for further large-scale efforts to clarify the divergent validity of the proposed marker.”

Page 17, line 13-14:

“Our confounder analysis found no evidence that the predictions of the proposed model are biased by variables of no interest ...”

Page 19, line 16-21:

“Limitations

(...)

Our study used a small discovery sample combined with pre-registered external validation. While this approach allows for an efficient and reliable construction of brain-based predictive models³⁷, our power calculation (Supplementary Figure 3) indicated that moderate to large samples⁴⁴ will be needed to thoroughly characterize the model in terms of its specificity to pain-related mechanisms, its relationship to other aspects of pain-related learning (e.g., resistance to extinction), and its validity in clinical populations.

”

Page 20, lines 11-13:

“Initial assessments of convergent and divergent validity indicate that the model may generalize to different measures of pain-related learning, but not to aversive tone, fear of pain and other potential confounders.”

Rev 2/Q6

6) It is excellent that you fix model weights and preregister. Looking at the correlation values between samples shows that performance in discovery model for RCPL is much better than in external samples (Rsquared difference 51.84 vs .11). The preregistration shows that elements of the models prediction are minimally accurate, but the performance drop off is quite alarming and suggests a non-trivial amount of overfitting in the model. At the end of the day, you all do much more than most to address overfitting by showing model performance after preregistration. I think the discussion would benefit from explicitly talking about why model performance drops off so much, how different models might perform etc.

Thank you for acknowledging the ability of our study to address overfitting. We agree that it is important to reiterate the reasons why performance estimates may be inflated, even in the case of proper internal validation, as in our study. In the revised manuscript we discuss this issue as follows:

Page 16 line 17-26:

“Of note, we observed a remarkable difference between internally and externally validated estimates of prediction performance. The reasons may be twofold. First, in small-sample model discovery, cross-validation is known to provide unbiased, but highly variable estimates^{38,39}. Second, in the case of complex preprocessing and feature engineering workflows, it is often not feasible to include all free parameters of the model in the nested cross-validation procedure^{37,38}. Seemingly ‘innocent’ adjustments of such workflows during model discovery can lead to biased internal performance estimates. Our study shows that both pitfalls can be overcome with pre-registered external validation; a technique that should become standard practice, in order to unlock small-sample model discovery and accelerate the discovery of robust and replicable multivariate brain-behavior associations^{43,44}.”

Rev 2/Q7

7) You should report the mean squared error (or root) for predictive performance. See PMID: 31774490 Russ Poldrack paper for why this is superior to showing correlation between predicted and observed values. I especially want to see mean squared error because the predicted and observed DVC shown in figure 1 seem to be quite different. The first tell of this is that you had to use different scales for the x and y axis in Fig1C. I am curious to see how close the model gets to actually predicting the observed DVC values. Importantly, Poldrack argues that variance explained is not appropriate for out-of-sample testing.

Following the Reviewer’s suggestion, we now report the root mean squared error in the manuscript.

Page 8, line 11-13:

“The root mean squared error values were 26.8, 25.03 and 26.9 for the merged sample, validation sample 1, and validation sample 2, respectively. The range of the observed values was [-14.5, 98.0] within the validation dataset.”

While the scales on *Figure 1* are different, this is a well-known effect of model regularization with Ridge regression. The phenomenon is widely discussed in the brain age literature (de Lange & Cole, 2020). As the mean squared errors show our model is relatively well-calibrated in external validation data, i.e., correctly predicts the mean of the population.

Rev 2/Q8

8) Figure 3A. What do the colors next to certain areas refer to? Are these networks? You should have some look up table or guide for these networks. The purple network looks somewhat alarming (if it is a network). It features a lot of areas with notoriously low SNR.

Yes, the colors denote large-scale networks as provided by the 7-region version of the BASC brain atlas (See https://simexp.github.io/multiscale_dashboard/index.html for an online visualization). Purple indicates the “mesolimbic” network. This information is now included in the caption of Figure 3:

Page 13 Figure 3 caption:

“Regions are grouped according to their parent network based on the seven parcellations of the BASC atlas. Red - somatomotor network; green – ventral attention+saliency network+basal ganglia + thalamus; blue – visual network; brown – cerebellar network; gray – default mode network; orange – fronto-parietal network+ visual downstream; purple – mesolimbic network.”

While some of the mesolimbic regions, such as the amygdala, do indeed often have low SNR (e.g., due to susceptibility artifacts), others, such as the posterior insula, are usually assessed with high quality. While we agree that low SNR can be problematic for fMRI-based markers, we believe that it should primarily lead to false negative (undetected) predictors, rather than false positives. Nevertheless, we have added the following note in the limitations section:

Page 19 line 25-28:

“Like any fMRI-based study, our study is also limited by low SNR in regions prone to artifacts (e.g. susceptibility artifacts), including regions that are known to be involved in pain-related learning, such as the orbitofrontal cortex. Results relating to these regions should therefore be interpreted with caution.”

Rev 2/Q9

9) The amygdala is notoriously challenging to localize with functional imaging. Based on the images provided, and the data driven parcellation used, I am somewhat skeptical how localized an ROI is to the amygdala. More anatomical data should be displayed to prove that you are actually looking at the amygdala. Can you also please confirm the amygdala peak (you report 3.06 -7.47 -19.9). I see you note that the center of mass can lie outside of the mass for bilateral parcels, in such cases, reporting the center mass is meaningless and somewhat confusing as it does not describe the ROI (maybe report left and right center mass). It might be available on your github, but would you be able to point to a .nii file containing the ROIs you used? You should provide additional information to demonstrate you are actually looking at the region you claim you are.

We agree with the Reviewer that localizing the amygdala can be challenging, especially in the case of the functional atlas we used, which is not constrained by *a priori* anatomical knowledge. The two amygdala regions in the BASC atlas are centered on the following coordinates: left: x=-23, y=-7, z=-20; right: x=24, y=-8, z=-20. Both are clearly located in the amygdala. We now provide an additional supplementary table with the correct information for all bilateral regions (*Supplementary Table 7*) and refer to it in *Table 3* caption. We also provide additional figures for the amygdala region in the Supplementary Materials (*Supplementary Figure 6*) and the atlas file in the project github page (https://github.com/kincsesbalint/paintone_rsn/tree/master/data_in/atlas_rois).

Page 12, Table 3 caption:

“Supplementary Table 7 contains MNI coordinates of left and right parcels separately for bilateral parcels.”

Supplementary Figure 6. The outline of the applied functional amygdala region in the coronal and axial slices of MNI space.

Note that the used atlas was based on functional parcellation, therefore the anatomical boundaries might be violated slightly. The used regions were back-projected to the individual functional space and intersected with the gray matter mask to minimize inaccuracies introduced by registration. The number of the initial and last slices in each orientation is shown, intermittent slices are sampled evenly. The location of each region can be found in Table 3 and Supplementary Table 7.

Rev 2/Q10

10) You should be defining the networks you are using in the discussion. What do you mean by parietal CEN?

We apologize for the lack of details and now define the 7 networks we refer to in the discussion (see R2/Q08). The central executive network (CEN), which is also known as the frontoparietal network (Smith et al., 2009), consists mainly of the dorsolateral prefrontal cortex and posterior parietal cortex, around the intraparietal sulcus. Our wording was intended to refer to our finding that the connectivity of a BASC region in the posterior parietal cortex with the amygdala was one of the most important predictors in our model and, while pointing out that this region functionally belongs to the CEN and is part of the frontoparietal network in the 7-region BASC parcellation. The text has been edited for clarity, (see changes in bold below):

Page 17, line 27:

*“Connectivity between the amygdala and the frontal part of the central executive network (CEN) (**which is also known as frontoparietal network**⁵⁹) is one of the most prevalent findings in aversive learning, ...”*

Page 17, line: 32 -Page 18, line 3:

“Our results add significantly these findings by highlighting the critical role of the connection between the amygdala and the parietal part of the CEN (occipitoparietal association area), a network widely recognised for its involvement in pain and perception more generally⁵⁹.”

Rev 2/Q11

11) Were the same $M=122$ regions used for all datasets? What dataset was used to generate the regions? How much would these regions change if they are estimated in a different dataset? Just curious, why did you choose to make your own parcellation instead of relying on a published parcellation?

We apologize for the lack of clarity in the original manuscript. The same, pre-defined parcellation was used in all studies (according to our preregistered pre-processing pipeline). This was the widely used functional parcellation derived using the BASC algorithm (Urchs et al., 2019), also known as MIST (Multiresolution Intrinsic Segmentation Template). We have now clarified this at multiple points in the main text and the methods (e.g., page 7 line 7, page 25 line 13-15).

“A functional brain atlas (data-driven parcellation derived with the BASC algorithm)⁴⁵ was used with 122 non-overlapping functional regions (the used atlas is available in the project github page https://github.com/kincsesbalint/paintone_rsn/tree/master/data_in) and additionally the global gray matter signal was also kept.”

Rev 2/Q12

12) Figure 3B. If you want to get a sense of dynamic range then you should pick a different color scaling. The difference between 10 and 5.2 is way too small to highlight dynamic range. If you want to just show all the areas that were significant, then just binarize the map.

We agree with the Reviewer and have reproduced the figure with a more appropriate surface sampling algorithm, to better show the differences in exclusion loss.

Figure 3 Resting-state functional connectivity predictors of pain-related learning.

A, Chord plot of all ten predictive connections between 20 regions. The sign of the β -coefficient is color coded, red (blue) indicates a positive (negative) sign, i.e., higher connections between the two regions leads to higher (lower) differential valence rating. The thickness of the chords is proportional to the absolute magnitude of the β -coefficient. Regions are grouped according to their parent network based on the seven parcellations of the BASC atlas. Red - somatomotor network; green – ventral attention+salience network+basal ganglia + thalamus; blue – visual network; brown – cerebellar network; gray – default mode network; orange – fronto-parietal network+ visual downstream; purple – mesolimbic network. **B**, Exclusion loss after leave-one-region-out (LORO). Leaving out some regions, such as the

amygdala or posterior insula leads to a large decrease in the explained variance of the model when tested in the external validation datasets. Only cortical regions with significant ($p < 0.05$) exclusion loss are visualized. **C**, Contrasting the association between the predictive connections' β -coefficients (the standard technique for assessing predictive performance) and exclusion loss (the proposed more robust measure of importance) in the leave-one-feature-out analysis, shows that 6 out of 10 predictive connectivities have little or no effect on the external model performance. However, the four most important connections, on the other hand, clearly stand out from the rest; excluding them during model training results in models that explain 6-10% less variance in the external samples. The red horizontal dashed line represents the significance threshold for the exclusion loss (5.8%). Abbreviations of the regions can be found in the caption of *Table 3*.

Rev 2/Q13

13) In the outlook section you talk about applying the RCPL to diverse populations. Can you give an example of how this would be applied to these populations and how this would be better than using behavioral data alone to calculate pain DVC? I would like to know how you see a signature like this being used.

We think this is a very important question and we are happy to share our view on this. Functional MRI-based predictive models can certainly be considered as direct substitutes or surrogate measures for subjective, self-reported target behavior. However, we would like to point out that the value of brain-based models goes far beyond this type of use. Because they are derived directly from neural data, such models can provide an objective measure of related processes and can explain variance in clinical variables over and above of that explained by subjective self-reports and behavioral measures. Motivated by this comment and Rev 3/Q1, we have performed a “sneak preview” analysis on chronic pain patients and found that the RCPL predicts chronic pain severity ($r = 0.51$, $p = 0.001$, $n = 37$; *Rebuttal Figure 1*, see also our response to Rev 3/Q1 and Supplementary Material *Model performance on the extended sample* for details). However, we believe that the validity, specificity, fairness, and robustness of our model need to be assessed in large, diverse samples before the model can be evaluated in clinical contexts. In the present work, we have focused on fulfilling two crucial prerequisites for such cumulative validation efforts: replicability (preregistered external validation) and accessibility (reproducible container). This already makes the RCPL a powerful “research tool”; a fixed network pattern that can serve as a basis for cumulative validation efforts, as well as an additional, well-defined rsfMRI outcome in studies about pain-related learning and chronic pain (see also our reply to Rev3/Q1 for initial analyses). The Neurologic Pain Signature (NPS) (Wager et al., 2013) may be one of the most straightforward examples for such cumulative research processes. After the initial discovery in a small sample ($n = 20$), the NPS has been tested in a series of studies with independent samples (for example, more than 600 participants from 20 independent studies in (Zunhammer et al., 2018)). This process has not only established the validity of the NPS in large and diverse samples, but has also informed the community about what it does (nociception, largely independent of sensory modality) and does not capture (e.g., aversive images, social rejection, placebo analgesia, cognitive reappraisal), thereby significantly advancing our understanding of the underlying neural mechanisms.

Rev 2/Q14

14) You say the cerebellum is known to be involved in fear condition behavior and that the cerebellum is a promising new target. This is bizarre to say. It is similar to saying, the cerebral cortex is known to be involved in pain and would be a promising target for research. Yes this is true, but it lacks any specificity. The cerebellum is enormous and has a similar surface area to the entire cerebral cortex.

We fully agree with the Reviewer that it is almost trivial to say that the cerebellum should be the subject of further investigation in pain-related learning. However, we believe that it is very useful to encourage further research into the cerebellum in our discussion, as cerebellar function is notoriously overlooked in similar studies. We have now rephrased the sentence as follows to better convey our message:

Page 18, line22-29:

“Finally, the cerebellum is known to be involved in fear conditioning behavior⁶⁸⁻⁷⁰ and there is now also accumulating evidence for a direct involvement of the cerebellum in pain modulation^{67,71}. However, the cerebellum is often overlooked in related research. Here we found that omitting cerebellar connectivity

significantly reduced predictive performance (~6% less variance explained). These results provide additional evidence that the commonly reported pain- and learning-associated cerebellar activity is not a secondary consequence of the experimental paradigms in these studies (e.g., suppression of motor responses) but it has a unique predictive contribution to the underlying mechanisms that is not redundant with activity and connectivity in other parts of the brain.”

Rev 2/Q15

15) I think the writing, especially in the introduction, does not do a good job of setting up a research question.

We have carefully reviewed the introduction and modified it in several places, including the following lines and phrased an explicit research question (Page 3, line 26-28):

Page 3, line 12-13:

“In acute pain, we learn to associate painful experiences with certain stimuli or situations, which helps us to adapt by learning to avoid or minimize future harm.”

Page 3, line 26-28:

“However, it has yet to be determined whether resting-state brain connectivity can characterize, and provide a better understanding of, individual differences in pain-related learning.”

Rev 2/Q16

16) The text for ‘preregistered’ used in some infographics is pixelated.

We thank the Reviewer for noticing this error and have updated the figure.

Rev 2/Q17

17) SFig1 has an odd partially cut off ‘2’ on the bottom left.

This has now been corrected.

Reviewer #3*In this study, Kincses et al. developed and evaluated a resting state functional brain connectivity based predictive marker of individual differences in pain-related learning, as measured by an established classical conditioning paradigm. For this they used 8 - 10 mins of resting state data collected prior to an established conditioning task to predict learning performance on the task, quantified as differential valence ratings (i.e., learning-induced increase in pain ratings; and sound ratings as a control condition for this study). They trained a machine learning model on whole brain data between 122 functionally defined regions to predict valence ratings reflecting pain related learning. The ML model was developed and tuned in the discovery sample (n = 25), then locked and preregistered, and then tested in 2 independent samples (n = 26 and n = 23) performing the equivalent conditioning paradigm.*

The model's predictive accuracy seems to be on par with several other efforts in genetic or brain marker development of predicting between-subject traits. The study results have a clear clinical relevance as a potential easily applied prognostic and diagnostic marker (requiring only resting state fMRI data) and neuroscientific merit by advancing our understanding of brain mechanisms of pain threat learning. The study implements several guardrails against ML-based issues (overfitting, effect of confounds, misinterpretation), which is commendable.

We are grateful to the Reviewer for acknowledging that the proposed model may have a clear clinical relevance, a competitive predictive performance and potential for advancing our understanding of brain mechanisms.

I have several reservations, as listed in the following.

Rev 3/Q1

1. Given the massive translational clinical relevance of this signature – as also emphasized by the authors in their introduction and discussion, as well as the ease of application to new datasets (“As the RCPL signature is derived solely from resting-state functional connectivity, and therefore requires only a 10-minute non-invasive scan without the use of noxious stimuli or time-consuming learning paradigms, it is a highly accessible tool for these clinical and translational purposes”) it seems like a missed opportunity not to have tested this signature in a publicly (or otherwise) available sample of people with chronic pain. Such a test – even if only as a preliminary validation in a relevant clinical sample - would likely strengthen the claims of the paper.

We thank the Reviewer for pointing out the high translational clinical relevance of our work and we could not agree more on the importance of testing the proposed model in those suffering from chronic pain. However, we ultimately refrained from doing so because we believe that it is still too early for this step. As we also discuss in our response to Rev 2/Q13, testing whether the reported neural signature is differently expressed in different clinical conditions needs to be preceded by a thorough evaluation of the robustness of our model to specific sample characteristics (e.g. altered in-scanner motion, confounders, fairness). However, motivated by the Reviewer's comment, we performed an additional analysis to provide further information on the robustness of our model to the exclusion of high-motion participants. We found that even when we completely dropped our pre-registered motion-exclusion criteria (~20% more individuals), the generalizability of our model remained unchanged ($r=0.35$,

p=0.0028, see Supplementary Material: *Model performance on the extended sample, Supplementary Figure 4*).

Furthermore, we performed an initial evaluation of our model in an openly available clinical dataset (OpenPain <https://www.openpain.org> – Brain Network Change, (Seymour et al., 2018)), and found that the signature response is strongly and significantly associated with chronic pain severity (Spearman correlation coefficient $r=0.51$, $p=0.001$, $n=37$; see *Rebuttal Figure 1*). While we consider this result to be promising evidence for the involvement of the reported network in chronic pain, we believe that a fair discussion of these findings will only be possible if the ability of our model to predict differential valence in chronic pain patients is first verified. We intend to acquire the data necessary for this validation step (including learning measures from compatible conditioning paradigm) in a separate project. Nevertheless, we have added the following note in the results section:

Rebuttal Figure 1. Correlation between RCPL score and chronic pain severity (self-reported scores on a 0-10 scale) in $n=37$ patients from the Brain Network Change dataset (Seymour et al., 2018), openly available at OpenPain.

Page 8, line 13-16:

“Interestingly, completely dropping our preregistered motion-exclusion (~20% more individuals, the performance of the model remained unchanged ($r=0.35$, $p=0.0028$, see also Supplementary Material: Model performance on the extended sample, Supplementary Figure 4).”

Rev 3/Q2

2. Re. specificity of the signature to predict pain vs. sound: I might not be understanding the Generalization test, but could not an alternative plausible explanation be that transfer learning (pain to sound) is a potential mechanism involved in aversive learning and therefore the results support a more general aversive learning? Some additional clarification as to why the presented result strongly favors pain-specific interpretation might be needed.

We apologize for the lack of detail on the partial generalization test in the original manuscript. Based on the Reviewer’s comment, as well as Rev 1/Q2-4 and Rev 2/Q4-5, we have significantly improved the presentation of this novel method (see also our responses to these comments for more detail).

To better assess the specificity of the proposed model, we have performed power calculations (see also Rev 1/Q4 and *Supplementary Figure 3* for details). These suggest that reliable and accurate disentangling of common and distinct mechanisms between pain and tone-related learning is only possible with large samples, and our findings on specificity must be interpreted with caution. We now acknowledge this as limitation in the revised manuscript.

We find the Reviewer’s comment on transfer learning very intriguing, and we fully agree that, pain- and tone-related learning may not only simply share common mechanisms, but may even interact with each other in our paradigm, through transfer learning and “cross-modality” generalization. We now mention this as an additional limitation in the revised manuscript:

Page 19, line 22-24:

“Disentangling the common and distinct mechanisms of pain- and tone-related learning is further limited in the present study by their potential interactions in our paradigm, through transfer learning and cross-modality generalization.”

Rev 3/Q3

3. Given how similar the conditioned ratings for pain and sound are (suppl. Fig 1) would it make sense to consider training on the sound data and examine whether a sound learning signature is specific to sound or also sensitive to pain to further isolate the pain-specificity of the developed signature?

We would like to thank the Reviewer for this suggestion. We fully agree that comparing models trained separately for pain and tone could provide valuable insight into common and distinct mechanisms. However, we found that the discovery sample lacked sufficient power to achieve significant predictive performance for aversive tone learning using nested cross-validation (see also our response to Rev 1/Q3 and Supplementary Material *Model training on tone related learning as target*). It is important to note that the lack of power does not affect the robustness of our findings about our model for pain-related learning, as these results were evaluated in a pre-registered external validation, following best practices for small sample model discovery (Gallitto et al., 2023; Poldrack et al., 2020).

In summary, although our current study design does not allow for the analysis suggested by the Reviewer, it serves as a basis for scaling up model discovery to adequately powered samples, allowing the investigation of shared and specific mechanisms. The revised paper now includes an expanded discussion of the limitations arising from the small discovery sample (see also Rev 1/Q6).

Rev 3/Q4

4. Re. number of subjects used in the paper, it is a bit confusing to keep seeing the size of the initial sample (before exclusions) reported ($n = 99$; for example in behavioral results table 1, and in Results first paragraph) when actually $n = 74$ were used in signature development and validation (after exclusions) across the 3 sub-samples (model development and internal cross-validation, external validations: $n = 25$; $n = 26$, $n = 23$) and for observed-predicted performance calculation and the corresponding scatter plots (Figure 1, Suppl. Figure 1). It might be more accurate /less confusing to primarily report $n = 74$ (and its sub-sample numbers).

We have revised the manuscript according to the Reviewer’s suggestion. *Table 1* has been updated (Page 7) to show post-exclusion numbers and the total sample table has been moved in the Supplementary Material (*Supplementary Table 4*). We have also adjusted the text in several places, for instance:

	PAIN		AVERSIVE TONE	
	Valence $\Delta(\Delta CS^+_{\text{pain}}, \Delta CS^-)$ (95% CI)	Contingency* CS^+_{pain} (95%CI)	Valence $\Delta(\Delta CS^+_{\text{tone}}, \Delta CS^-)$ (95%CI)	Contingency* CS^+_{tone} (95%CI)
Discovery Study (n=25)	25 (16-34)	56 (40-70)	15 (7-25)	22 (1-42)
Validation Study (n=49)				
sub-sample 1 (n=26)	21 (12-31)	48 (29-65)	15 (4-25)	19 (1-37)
sub-sample 2 (n=23)	29 (18-41)	66 (50-80)	27 (16-38)	70 (55-84)

Table 1 Behavioral results

Page 3, line 3-4:

“We assessed individual learning performance related to pain and aversive tone using a differential conditioning paradigm (**Supplementary Figure 1**) in a total of $n=99$ ($n=74$, after exclusion)…”

Page 7, line 2-4:

*“Resting-state functional MRI data were collected from all 99 participants (**n=74, after exclusion**) in the three studies for a duration of eight to ten minutes per participant, prior to the behavioral experiments (see Supplementary Table 1).”*

In addition, when discussing exclusions in the manuscript, we now refer to our analyses without exclusions (Supplementary Material *Model performance on the extended sample*), to demonstrate that our results hold independently of exclusions.

Page 8, line 13-16:

“Interestingly, completely dropping our preregistered motion-exclusion (~20% more individuals, the performance of the model remained unchanged ($r=0.35$, $p=0.0028$, see also Supplementary Material: Model performance on the extended sample, Supplementary Figure 4).”

- de Lange, A. M. G., & Cole, J. H. (2020). Commentary: Correction procedures in brain-age prediction. In *NeuroImage: Clinical* (Vol. 26). <https://doi.org/10.1016/j.nicl.2020.102229>
- Forkmann, K., Wiech, K., Schmidt, K., Schmid-Köhler, J., & Bingel, U. (2023). Neural underpinnings of preferential pain learning and the modulatory role of fear. *Cerebral Cortex*. <https://doi.org/10.1093/cercor/bhad236>
- Gallitto, G., Englert, R., Kincses, B., Kotikalapudi, R., Hoffschlag, K., Bingel, U., Büchel, C., & Spisak, T. (2023). External validation of machine learning models with adaptive sample splitting. *BioRxiv*.
- Koenen, L. R., Icenhour, A., Forkmann, K., Theysohn, N., Forsting, M., Bingel, U., & Elsenbruch, S. (2018). From Anticipation to the Experience of Pain: The Importance of Visceral Versus Somatic Pain Modality in Neural and Behavioral Responses to Pain-Predictive Cues. *Psychosomatic Medicine*, 80(9). <https://doi.org/10.1097/PSY.0000000000000612>
- Korn, E. L. (1984). The ranges of limiting values of some partial correlations under conditional independence. *American Statistician*, 38(1). <https://doi.org/10.1080/00031305.1984.10482876>
- Lonsdorf, T. B., & Merz, C. J. (2017). More than just noise: Inter-individual differences in fear acquisition, extinction and return of fear in humans - Biological, experiential, temperamental factors, and methodological pitfalls. *Neuroscience and Biobehavioral Reviews*, 80. <https://doi.org/10.1016/j.neubiorev.2017.07.007>
- Poldrack, R. A., Huckins, G., & Varoquaux, G. (2020). Establishment of Best Practices for Evidence for Prediction: A Review. In *JAMA Psychiatry* (Vol. 77, Issue 5). <https://doi.org/10.1001/jamapsychiatry.2019.3671>
- Schlitt, F., Schmidt, K., Merz, C. J., Wolf, O. T., Kleine-Borgmann, J., Elsenbruch, S., Wiech, K., Forkmann, K., & Bingel, U. (2021). *Impaired pain-related threat and safety learning in patients with chronic back pain*. <https://doi.org/10.1097/j.pain.0000000000002544>
- Seymour, B., Mano, H., Kotecha, G., Leibnitz, K., Matsubara, T., Nakae, A., Shenker, N., Shibata, M., Voon, V., Yoshida, W., Lee, M., Yanagida, T., Kawato, M., & Rosa, M. J. (2018). Classification and characterisation of brain network changes in chronic back pain: A multicenter study. *Wellcome Open Research*, 3. <https://doi.org/10.12688/wellcomeopenres.14069.1>
- Smith, S. M., Fox, P. T., Miller, K. L., Glahn, D. C., Fox, P. M., Mackay, C. E., Filippini, N., Watkins, K. E., Toro, R., Laird, A. R., & Beckmann, C. F. (2009). Correspondence of the brain's functional architecture during activation and rest. *Proceedings of the National Academy of Sciences of the United States of America*, 106(31). <https://doi.org/10.1073/pnas.0905267106>
- Spisak, T. (2022). Statistical quantification of confounding bias in machine learning models. *GigaScience*, 11. <https://doi.org/10.1093/gigascience/giac082>
- Spisak, T., Bingel, U., & Wager, T. D. (2023). Multivariate BWAS can be replicable with moderate sample sizes. *Nature*, 615(7951). <https://doi.org/10.1038/s41586-023-05745-x>
- Varoquaux, G. (2018). Cross-validation failure: Small sample sizes lead to large error bars. *NeuroImage*, 180. <https://doi.org/10.1016/j.neuroimage.2017.06.061>
- Wager, T. D., Atlas, L. Y., Lindquist, M. A., Roy, M., Woo, C.-W., & Kross, E. (2013). An fMRI-Based Neurologic Signature of Physical Pain. *New England Journal of Medicine*, 368(15). <https://doi.org/10.1056/nejmoa1204471>

Zlomuzica, A., Preusser, F., Schneider, S., & Margraf, J. (2015). Increased perceived self-efficacy facilitates the extinction of fear in healthy participants. *Frontiers in Behavioral Neuroscience, 9*(OCTOBER). <https://doi.org/10.3389/fnbeh.2015.00270>

Zunhammer, M., Bingel, U., & Wager, T. D. (2018). Placebo Effects on the Neurologic Pain Signature: A Meta-analysis of Individual Participant Functional Magnetic Resonance Imaging Data. *JAMA Neurology, 75*(11). <https://doi.org/10.1001/jamaneurol.2018.2017>

REVIEWERS' COMMENTS:

Reviewer #1 (Remarks to the Author):

The authors have addressed most of my concerns. I have only two minor comments:

1. The terms "generalizability", "specificity", and "confounding" were not defined as the ML community generally uses them (although a complete consensus may not be available). The authors cited their previous peer-reviewed paper to argue for their criteria for specificity, generalizability, and confounding, but their approach is rather new and not generally accepted yet. To avoid any confusion and misinterpretation of the present paper, the authors can discuss a bit more about the potential differences between their terms and others, and remind readers of the differences. If possible, I think it's also better for the authors to test the generalizability, specificity, and confounding bias according to the common criteria.

2. Results about the power analysis are mentioned only in the Discussion section. It's better to move them to the Results section.

Reviewer #3 (Remarks to the Author):

My concerns have been addressed and I also find the responses to the other reviewer's concerns satisfactory.

My only remaining suggestion (no need to send it back to me for review) would be to expand the "Model training on tone related learning as target" by summarizing briefly the output of the github script (=lack of sig. predictions) and adding the helpful explanation from the Rev1/Q3 "While our pre-registered external validation is well positioned to confirm our true positive discoveries (predictive model of pain-related learning), it is not able to confirm whether the lack of significant predictions for tone-related learning is a false negative observation (Gallitto et al., 2023)" to provide a bit more context for why "the lack of power does not affect the robustness of our findings about our model for pain-related learning, as those results were evaluated in a pre-registered external validation, following the best practices for small sample model discovery (Gallitto et al., 2023; Poldrack et al., 2020)."

Point-by-point response to reviewers' comments

Reviewer #1

The authors have addressed most of my concerns. I have only two minor comments:

Rev 1/Q1:

1. *The terms “generalizability”, “specificity”, and “confounding” were not defined as the ML community generally uses them (although a complete consensus may not be available). The authors cited their previous peer-reviewed paper to argue for their criteria for specificity, generalizability, and confounding, but their approach is rather new and not generally accepted yet. To avoid any confusion and misinterpretation of the present paper, the authors can discuss a bit more about the potential differences between their terms and others, and remind readers of the differences. If possible, I think it’s also better for the authors to test the generalizability, specificity, and confounding bias according to the common criteria.*

We agree that would be important to avoid confusion or misinterpretation of the used terms. We now explicitly distinguish the defined terms from other possible interpretations. We have added the following section:

Page 7:

“The concepts of convergent and divergent validity are not yet fully developed in the field of machine learning. Importantly, the term generalization (to other measures of pain or convergent validity) should not be confused with the model generalization to new, unseen data (i.e.: external validity see above). Additionally, our specificity analysis (divergent validity) tests whether the model captures aversive learning (tone-related) in general and differs from the similar term used in classification problems.”

2. Results about the power analysis are mentioned only in the Discussion section. It’s better to move them to the Results section.

Following the Reviewer’s suggestion, we moved the paragraph below from the *Discussion* to the *Results* section.

Page 8:

“While our power analysis (Supplementary Figure 5) suggests that larger samples are needed to reliably disentangle general and pain-specific mechanisms in aversive learning, our results provide a promising basis for further large-scale efforts to clarify the divergent validity of the proposed marker. “

Reviewer #3

My concerns have been addressed and I also find the responses to the other reviewer’s concerns satisfactory.

1. *My only remaining suggestion (no need to send it back to me for review) would be to expand the “Model training on tone related learning as target” by summarizing briefly the output of the github script (=lack of sig. predictions) and adding the helpful explanation from the Rev1/Q3*

“While our pre-registered external validation is well positioned to confirm our true positive discoveries (predictive model of pain-related learning), it is not able to confirm whether the lack of significant predictions for tone-related learning is a false negative observation (Gallitto

et al., 2023)"

to provide a bit more context for why "the lack of power does not affect the robustness of our findings about our model for pain-related learning, as those results were evaluated in a pre-registered external validation, following the best practices for small sample model discovery (Gallitto et al., 2023; Poldrack et al., 2020)."

We thank the Reviewer for this suggestion. We implemented the recommended changes in the *Supplementary: Model training on tone related learning as target*.

"That is, the result of cross validation procedure was non-conclusive and the fitted model was not capable to predict tone related learning in the validation datasets (see scatter plots of validation dataset 1,2 in the github page). It is important to note that the lack of power does not affect the robustness of our findings about our model for pain-related learning, as those results were evaluated in a pre-registered external validation, following the best practices for small sample model discovery^{4,5}. While our pre-registered external validation is well positioned to confirm our true positive discoveries (predictive model of pain-related learning), it is not able to confirm whether the lack of significant predictions for tone-related learning is a false negative observation⁵"